# Unraveling and leveraging in situ surface amorphization for enhanced hydrogen evolution reaction in alkaline media

Qiang Fu[1,7], Lok Wing Wong[1,7], Fangyuan Zheng [1], Xiaodong Zheng[1], Chi Shing Tsang[1], Ka Hei Lai[1], Wenqian Shen[1], Thuc Hue Ly [2,3,4] ✉, Qingming Deng [5] ✉ & Jiong Zhao [1,6] ✉

Surface amorphization provides electrocatalysts with more active sites and flexibility. However, there is still a lack of experimental observations and mechanistic explanations for the in situ amorphization process and its crucial role. Herein, we propose the concept that by in situ reconstructed amorphous surface, metal phosphorus trichalcogenides could intrinsically offer better catalytic performance for the alkaline hydrogen production. Trace Ru (0.81 wt.%) is doped into $NiPS_3$ nanosheets for alkaline hydrogen production. Using in situ electrochemical transmission electron microscopy technique, we confirmed the amorphization process occurred on the edges of $NiPS_3$ is critical for achieving superior activity. Comprehensive characterizations and theoretical calculations reveal Ru primarily stabilized at edges of $NiPS_3$ through in situ formed amorphous layer containing bridging $S_2^{2-}$ species, which can effectively reduce the reaction energy barrier. This work emphasizes the critical role of in situ formed active layer and suggests its potential for optimizing catalytic activities of electrocatalysts.

Electrochemical water splitting is a significant hydrogen production technique and involves a complex surface chemical reaction[1,2]. The catalytic activity is highly dependent on the surface structure of the catalytic materials, particularly in the case of alkaline water electrolysis (AWE)[3–5]. For the alkaline HER process, surface reconstruction is a frequently observed phenomenon and usually leads to the formation of an amorphous layer outside the catalysts[6,7]. However, the fundamental understanding of the surface amorphization process and the role of the resulting amorphous layer on catalysts is still deficient. Additionally, how to effectively leverage the inevitable amorphous layer to further improve the catalytic activity of electrocatalytic material remains a challenging task[8]. Hence, it is imperative to identify

a proper material platform to comprehensively investigate the impact of the amorphous layer on the catalytic activity.

Recently, two-dimensional (2D) metal phosphorus trichalcogenides (MPTs) have garnered increasing attention as catalysts for the hydrogen evolution reaction (HER)[9–13]. Compared with other reported transition metal-based electrocatalysts, 2D MPTs demonstrate unique crystal structures with a metal layer encapsulated by both chalcogens and phosphorus atoms, as well as abundant $[P_2S_6]^{4-}$ functional groups. These features provide high specific surface area, tunable charge states, and appropriate band structures[12,14–16]. Due to the semiconductor nature and inert basal plane of MPTs, the catalytic active sites are mainly concentrated at the edge positions[13]. Therefore, using

[1]Department of Applied Physics, The Hong Kong Polytechnic University, Kowloon, China. [2]Department of Chemistry and Center of Super-Diamond & Advanced Films (COSDAF), City University of Hong Kong, Kowloon, China. [3]Department of Chemistry and State Key Laboratory of Marine Pollution, City University of Hong Kong, Hong Kong, China. [4]City University of Hong Kong Shenzhen Research Institute, Shenzhen, China. [5]Phyics Department and Jiangsu Key Laboratory for Chemistry of Low-Dimensional Materials, Huaiyin Normal University, Huaian, China. [6]The Hong Kong Polytechnic University Shenzhen Research Institute, Shenzhen, China. [7]These authors contributed equally: Qiang Fu, Lok Wing Wong. ✉e-mail: thuchly@cityu.edu.hk; qingmingdeng@gmail.com; jiongzhao@polyu.edu.hk

2D MPTs as a research subject can effectively mitigate the influence of extraneous factors, enabling more precise investigations into the effects of the amorphous layer on catalytic sites and the overall activity of the material.

Herein, we introduce an edge optimization strategy to enhance the catalytic activity of MPTs. We leverage the inevitable surface reconstruction that occurs during the alkaline HER process to stabilize dopants (Ru) in the in situ formed amorphous layer, thereby enhancing the adsorption ability for intermediates, and increasing the number of active sites. The modified Ru-NiPS$_3$ nanosheets (NSs) demonstrated an overpotential ($\eta_{10}$) of 58 mV to reach the current density of 10 mA cm$^{-2}$ and a high exchange current density of 1180 μA cm$^{-2}$, which is comparable to the commercial Pt/C catalyst. Comprehensive characterizations and density functional theory (DFT) calculations revealed that the Ru atoms were trapped by the in situ formed bridging S$_2^{2-}$ species in the amorphous layer. This reduces the leaching of Ru dopant and modifies the electronic structure around the active sites. These findings confirm the benefits of in situ surface amorphization in transition metal-based electrocatalysts and provide an alternative avenue for designing highly efficient catalysts for electrochemical applications.

## Results and discussion
### Synthesis and structural characterization

Ru-NiPS$_3$ NSs electrode was prepared through a three-step procedure (as shown in Supplementary Fig. 1). The obtained Ni precursor was first characterized with X-ray diffraction (PXRD) and scanning electron microscopy (SEM) image (Supplementary Figs. 2, 3), which demonstrated NSs morphology. For comparison, single crystal NiPS$_3$ as a control sample was also prepared with a CVT method (details can be found in the experimental section in Supporting Information).

We thoroughly examined the morphology and crystal structure of our prepared samples. The SEM images demonstrated that the Ru-NiPS$_3$ nanosheets had grown uniformly in a hexagonal shape on the surface of carbon cloth with an average thickness of about 150 nm (Fig. 1a and Supplementary Figs. 4, 5). We also characterized the morphology and crystal structure of NiPS$_3$ NSs without Ru doping for comparison and found that they were similar in shape to the Ru-NiPS$_3$ nanosheets. (Supplementary Figs. 6, 7a). To confirm the quality of our NiPS$_3$ nanosheets, we further analyzed their energy-dispersive X-ray spectroscopy spectra, which showed a stoichiometric ratio of Ni:P:S = 1:1:3 (Supplementary Fig. 7b).

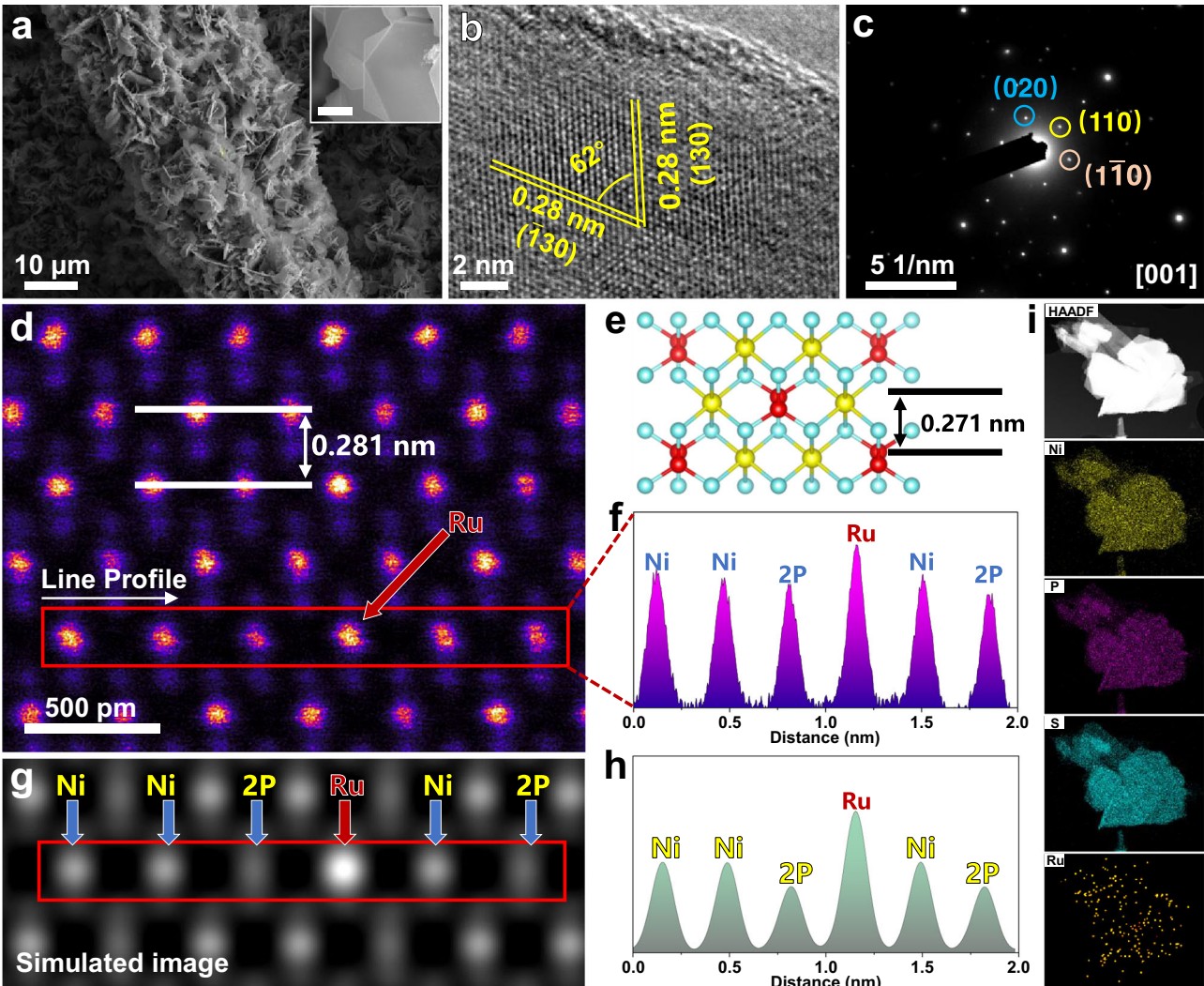

**Fig. 1 | Structural characterizations of Ru-NiPS$_3$ NSs. a** SEM image of the as-prepared Ru-NiPS$_3$ NSs. **b, c** HRTEM image and the corresponding SAED image along the [001] zone axis. **d** Atomic-level HAADF-STEM image of an ultrathin NiPS$_3$ nanosheet. **e** Crystal structure of NiPS$_3$ along [001] zone axis. **f** Line intensity profile obtained from the selected area in (**d**). (**g**) and (**h**) are the simulated HAADF-STEM image with Ru doped into the NiPS$_3$ lattice and the corresponding line intensity profile which is similar to the experimental one. **i** HDDF-STEM image corresponding EDS mapping of Ru-NiPS$_3$ nanosheet (scale bar 500 nm).

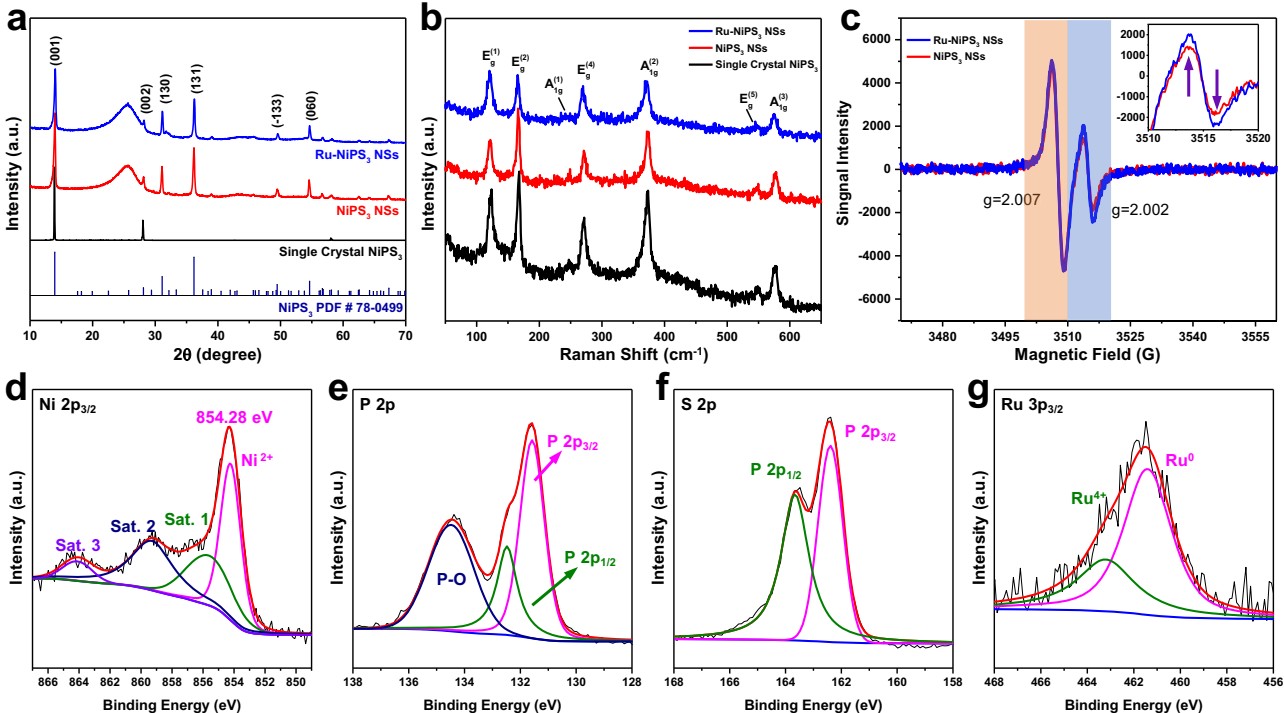

**Fig. 2 | Compositional and electronic structure of as-prepared electrocatalysts. a** PXRD patterns of Single crystal $NiPS_3$ powder, pure $NiPS_3$ NSs, and Ru-$NiPS_3$ NSs. **b** Raman spectra of the samples as mentioned above. **c** EPR results of $NiPS_3$ and Ru-$NPS_3$. XPS spectra of (**d**) Ni $2p_{3/2}$, (**e**) P 2p, (**f**) S 2p, and (**g**) Ru 3p.

Samples that were immersed in $RuCl_3$ solution for varying durations demonstrated comparable morphology to the Ru-NiPS3 nanosheets described earlier (Supplementary Figs. 8–11). The high-resolution transmission electron microscopy (HRTEM) image of Ru-$NiPS_3$ NSs showed two different interplanar distances of 0.286 nm and 0.280 nm with a dihedral angle of 62°, corresponding to the (130) and ($\bar{1}$30) planes (Fig. 1b)[17]. The corresponding selected area electron diffraction (SAED) pattern was indexed to the monoclinic crystal structure along the [001] zone axis (Fig. 1c), which closely matched the theoretical $NiPS_3$ crystal. To determine the precise distribution of Ru atoms within the nanosheets, aberration-corrected high-angle annular dark-field scanning transmission electron microscopy (AC HAADF-STEM) imaging characterization was conducted (Fig. 1d). The AC HAADF-STEM image was taken along the [001] zone axis, with an adjacent lattice fringe of 0.281 nm, which matched well with the theoretical $NiPS_3$ crystal (Fig. 1e). By analyzing the intensity profile of the nanosheets, it was confirmed that the Ru atoms were doped into the $NiPS_3$ lattice site, occupying the same position as the Ni atoms (Fig. 1f), which is consistent with the simulated HAADF image and the corresponding intensity profile (Fig. 1g, h).

To further verify the position of the Ru atoms within the nanosheets, we also obtained the AC HAADF-STEM image along the [013] zone axis, which confirmed that the Ru atoms had replaced the Ni atoms during the ion exchange process (Supplementary Fig. 12). At the same time, STEM-EDS mapping showed clearly that Ni, P, S, and Ru elements were uniformly distributed within the nanosheet (Fig. 1i). Finally, we used inductively coupled plasma-optical emission spectroscopy (ICP-OES) analysis to confirm the amount of Ru in the nanosheets and it revealed that the Ru content was 0.81 wt.% (Supplementary Table 1). The ICP-OES results of other samples treated with different dipping durations were demonstrated in (Supplementary Table 2), which indicated that the maximum loading of Ru species by ion exchange methods is about 0.8 wt.%.

To gain further insights into the structure and chemical states of the prepared samples, we used several other analytical techniques, including powder X-ray diffraction (PXRD), Raman spectroscopy, and X-ray photoelectron spectroscopy (XPS). As shown in Fig. 2a, the single crystalline $NiPS_3$ showed strong diffraction peak intensity of (001), (002) and (004) which belonged to $NiPS_3$ (PDF #78-0499, space group: $C2/m$, and unit cell parameters: $a = 5.812$ Å, $b = 10.070$ Å, and $c = 6.632$ Å), indicating the high quality of our prepared sample. Meanwhile, the PXRD patterns for both the $NiPS_3$ nanosheets and the Ru-doped $NiPS_3$ nanosheets revealed their polycrystalline structure and the varying immersing durations during sample preparation will not change the crystal structure of $NiPS_3$ (Fig. 2a and Supplementary Fig. 13).

The Raman spectra of different samples demonstrated four in-plane $E_g$ and three out-of-plane $A_{1g}$ vibrational modes within the range of $100–600$ cm$^{-1}$. The peak below 150 cm$^{-1}$ was attributed to the $Ni^{2+}$ metal ion vibrations in $NiPS_3$, while other peaks below 600 cm$^{-1}$ were associated with the vibrational modes of the $PS_3$ group and P-P bond (Fig. 2b and Supplementary Fig. 14)[18]. The electron paramagnetic resonance (EPR) spectra of $NiPS_3$ and Ru-$NiPS_3$ were performed to study the concentration of vacancies. Two different symmetric EPR peaks were detected at $g = 2.002$ and 2.007, which were attributed to the presence of P vacancies (P-V)[19] and S vacancies (S-V)[18], respectively. The intensity of the P-V peak was similar in both samples, as revealed by the EPR spectra. However, the Ru-doped nanosheets exhibited a slightly higher intensity of the S-V peak compared to the pristine one, suggesting a higher concentration of S-V in Ru-$NiPS_3$ NSs (Fig. 2c). Usually, a higher concentration of S vacancy would provide more active sites and result in better HER activity[20]. As observed in the AC HAADF-STEM image, Ru atoms would replace Ni atoms after ion exchange. Considering the crystal structure characteristics (a Ni/P atom would coordinate with 6S atoms), the increment of S-V further indicated Ru would occupy Ni sites when doped into the lattice.

X-ray photoelectron spectroscopy (XPS) was performed to understand the surface chemical state of the Ru-$NiPS_3$ electrode (survey spectrum was demonstrated in Supplementary Fig. 15). The Ni $2p_{3/2}$ spectrum was deconvoluted into four peaks, indicating the

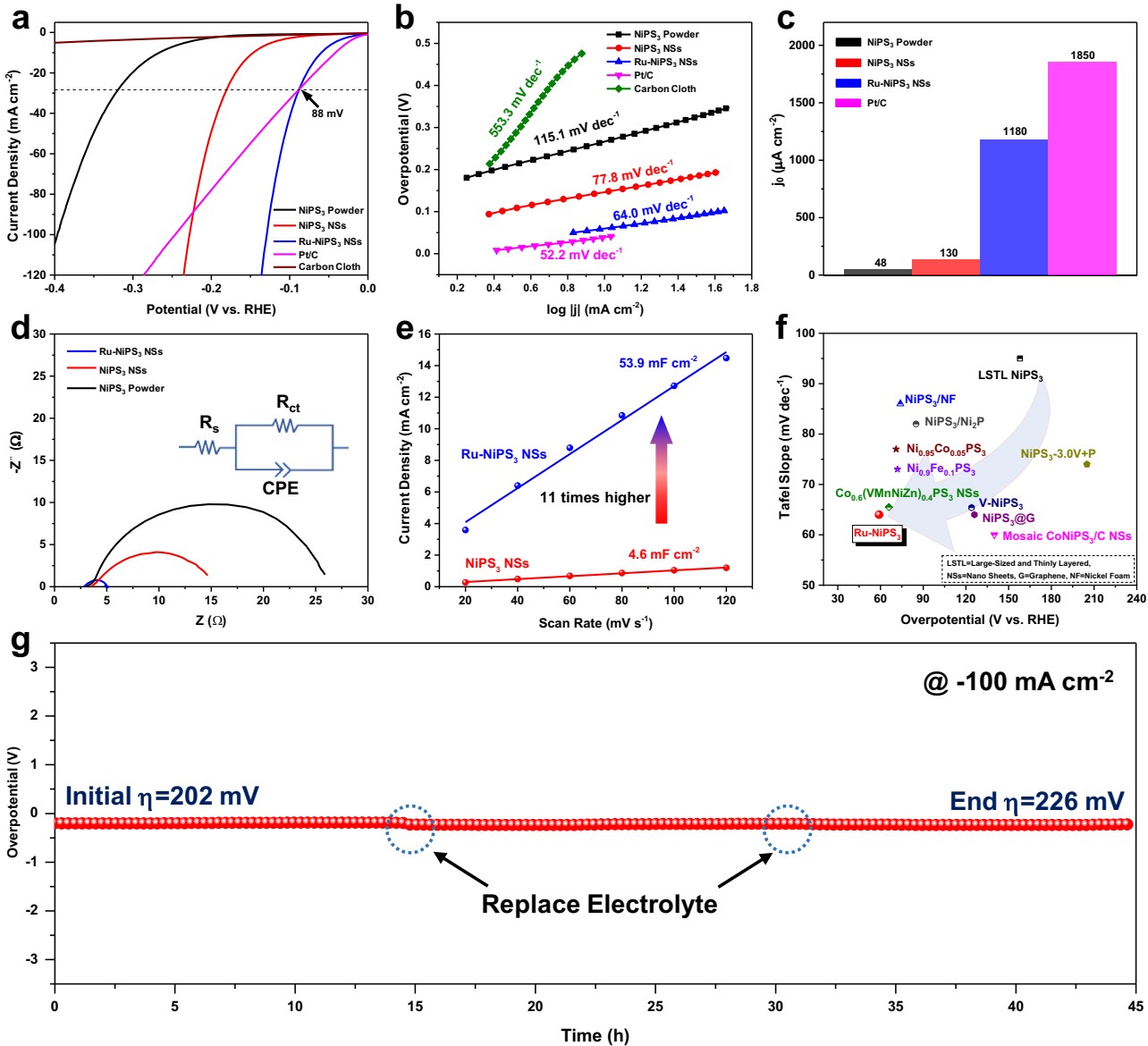

**Fig. 3 | HER performances of different samples in 1 M KOH (pH = 14). a** HER polarization curves for carbon cloth, NiPS$_3$ powder, NiPS$_3$ NSs, Ru-NiPS$_3$ NSs, and Pt/C measured in 1 M KOH with a scan rate of 2 mV s$^{-1}$. The mass loading of the electrocatalysts is -1.5 mg cm$^{-2}$, and the solution resistance is -3.0 Ω. **b** Tafel slope of different catalysts obtained from the polarization curves in (**a**). **c** Exchange current density of different samples extrapolated linearly from the Tafel slope in (**b**).

**d** Nyquist plots of NiPS$_3$ powder, NiPS$_3$ NSs, and Ru-NiPS$_3$ NS (inset shows the equivalent circuit diagram). (**e**) Scan rate dependence of the average capacitive currents for NiPS$_3$ NSs and Ru-NiPS$_3$ NSs. **f** Comparison of the overpotential and Tafel slope among our catalyst and other reported NiPS3-based electrocatalysts for alkaline HER. **g** Chronoamperometry curve test for Ru-NiPS$_3$ NSs at a fixed current density of −100 mA cm$^{-2}$.

presence of core levels of Ni$^{2+}$ (854.3 eV) and three corresponding satellite peaks located at 855.9 eV, 859.4 eV, and 864.2 eV, respectively[21,22]. The formation of NiPS$_3$ was further supported by the observation of three clear satellite peaks, which indicate the hybridization of Ni$^{2+}$ level and PS$_3$ ligand orbitals (Fig. 2d)[13]. The P 2p spectrum exhibited two distinct peaks at 131.6 eV and 132.5 eV, corresponding to P 2p$_{3/2}$ and P 2p$_{1/2}$, respectively, providing evidence for the formation of covalent P−S bonds within the PS$_3$ units. Furthermore, a broad peak at 134.5 eV was observed, likely arising from surface oxidation during the synthesis of transition metal-based phosphides (Fig. 2e)[23]. The S 2p spectrum also demonstrated two peaks at 162.4 eV and 163.7 eV, which belonged to the S 2p$_{3/2}$ and S 2p$_{1/2}$ spin−orbit peaks, respectively (Fig. 2f)[24]. Ru 3p$_{3/2}$ spectrum exhibited two deconvoluted peaks at 461.4 eV and 463.2 eV, which could be ascribed to Ru$^0$ and Ru$^{4+}$ [25]. The XPS spectra for undoped NiPS$_3$ NSs electrodes were also demonstrated for better

comparisons. The XPS spectrum of Ni 2p$_{3/2}$ and P 2p demonstrated similar deconvoluted peaks as the Ru-NiPS$_3$ sample without obvious peak shift (Supplementary Fig. 16a, b). However, the binding energy for S 2p$_{3/2}$ (162.2 eV) and S 2p$_{1/2}$ (163.4 eV) were lower than the Ru-NiPS$_3$ NSs, indicating the formation of Ru-S bond, and more electrons accumulated around Ru atoms due to its high electronegativity (Ru (2.2) and Ni (1.9)) (Supplementary Fig. 16c)[26].

## Electrocatalytic performances of Ru-NiPS$_3$ NSs

We investigated the hydrogen evolution reaction (HER) activity of Ru-doped NiPS$_3$ in 1 M KOH electrolyte. The electrode that had been dipped in RuCl$_3$ solution for 16 h demonstrated the most effective catalytic activity and was chosen as the representative sample (Supplementary Figs. 17, 18). As shown in Fig. 3a, Ru-NiPS$_3$ NSs exhibited a relatively low $\eta_{10}$ of 59 mV to achieve a current density of 10 mA cm$^{-2}$, which is significantly lower than the undoped NiPS$_3$ NSs ($\eta_{10}$ = 146 mV).

Moreover, the HER activity of Ru-NiPS$_3$ NSs was comparable to the commercial Pt/C electrocatalysts ($\eta_{10}$ = 41mV) and surpassed the latter when the overpotential reached 88 mV. Due to the low specific area and less exposed active sites, NiPS$_3$ powder showed the lowest HER performance with an $\eta_{10}$ of 266 mV to achieve current densities of 10 mA cm$^{-2}$. Tafel plots derived from the polarization curves showed that Ru-NiPS$_3$ NSs have the lowest Tafel slope of 64.0 mV dec$^{-1}$, which is much lower than the NiPS$_3$ NSs (77.8 mV dec$^{-1}$) and NiPS$_3$ powder (115.1 mV dec$^{-1}$), indicating a faster HER kinetics process through Volmer–Heyrovsky mechanism (Fig. 3b)[27,28]. The exchange current density for each sample was obtained using the Tafel plot extrapolation method. Ru-NiPS$_3$ NSs exhibited a current density of 1180 μA cm$^{-2}$, which is much larger than NiPS$_3$ NSs (130 μA cm$^{-2}$) and NiPS$_3$ powder (48 μA cm$^{-2}$), further indicating an enhanced intrinsic HER activity for Ru-NiPS$_3$ NSs (Fig. 3c)[2].

The HER kinetics for each sample was then investigated by electrochemical impedance spectroscopy (EIS). The Ru-NiPS$_3$ NSs demonstrated a lower charge transfer resistance ($R_{ct}$) compared to NiPS$_3$ powder and NiPS$_3$ NSs (Fig. 3d and Supplementary Table 3), suggesting that Ru-NiPS$_3$ NSs may have faster charge transfer and ultimately improved HER activities. Electrochemical double-layer capacitance ($C_{dl}$) was then utilized to estimate the electrochemical surface area (ECSA) and the concentration of catalytic active sites in each sample (Supplementary Figs. 19, 20). As shown in Fig. 3e and Fig. 18d, the value of $C_{dl}$ would gradually increase with increasing dipping duration. The Ru-NiPS$_3$ NSs demonstrated a $C_{dl}$ of 53.9 mF cm$^{-2}$, which is 11 times higher than that of the undoped NiPS$_3$ NSs (4.6 mF cm$^{-2}$), indicating the presence of a greater number of active sites available for HER.

Notably, the prepared Ru-NiPS$_3$ NSs demonstrated superior HER performance compared to most of the previously reported NiPS$_3$-based electrocatalysts (a summary of the data is presented in Fig. 3f and the data is extracted from Supplementary Table 4). The stability of the Ru-doped NiPS3 NSs electrode was assessed using the chronopotentiometry technique at a constant current density of 100 mA cm$^{-2}$. Impressively, the electrode demonstrated only a minor 24 mV increase in overpotential after continuous testing for 45 h, indicating its remarkable durability (Fig. 3g).

Temperature-dependent kinetic analysis was conducted to further study the origin of the enhanced catalytic activities[29]. LSV curves for Ru-NiPS$_3$ NSs and NiPS$_3$ NSs were obtained under different temperatures (from 25 °C to 65 °C) for the calculation of the activation energy ($E_{app}$) and pre-exponential factor ($A_{app}$), which are two essential parameters to evaluate the catalytic mechanism. As expected, the HER performances of both Ru-NiPS$_3$ NSs and NiPS$_3$ NSs were improved with increasing temperature (Supplementary Fig. 21). $E_{app}$ and $A_{app}$ were then calculated according to the Arrhenius equation by fitting the Arrhenius curve (Supplementary Fig. 22). It was found that the maximum value of $E_{app}$ for NiPS$_3$ NSs appeared around its onset potential, while the maximum value for Ru-NiPS$_3$ was at a relatively high overpotential. For the HER process in alkaline electrolytes, the rate-determining step (RDS) was the Volmer step, i.e., the water dissociation step due to the low concentration of H$^+$ in alkaline media, which consisted of the situation for NiPS$_3$ NSs[30]. However, the value of $E_{app}$ for Ru-NiPS$_3$ at the onset potential was much lower than its maximum at around 120 mV, indicating the RDS was no longer the Volmer step, and the desorption of H on the catalyst surface may be the limiting step for Ru-NiPS$_3$ (Supplementary Fig. 23a)[31]. The value of $A_{app}$ for Ru-NiPS$_3$ is higher than that of NiPS$_3$ NSs, suggesting a larger number of active sites participating in the HER (Supplementary Fig. 23b)[32–34]. The comprehensive electrochemical characterizations demonstrated that the incorporation of Ru into NiPS$_3$ NSs substantially modifies the reaction path, leading to a reduction in the kinetic barrier for the formation of intermediates during the Volmer step in alkaline solutions[5].

## Observation and characterization of surface reconstruction process

In situ liquid electrochemical TEM technique was conducted to gain insights into the structural evolution of Ru-NiPS$_3$ NSs during the alkaline HER process (Fig. 4a demonstrated the structure of the in situ electrochemical liquid cell TEM holder and the liquid cell). To minimize interference from other factors in the analysis, we first ruled out two potential sources of disturbance: corrosion of the sample by the alkaline solution, and interference from the electron beam on the sample. The Ru-NiPS$_3$ NSs electrode was first immersed into 1 M KOH for 24 h to show the influence of the alkaline electrolyte. As demonstrated in Supplementary Fig. 24a, the morphology demonstrated negligible variation after the immersing process, which is also proved by the corresponding XRD pattern (Supplementary Fig. 24b) and Raman spectra (Supplementary Fig. 24c). To rule out the influence of the electron beam in the TEM, the assembled liquid in situ electrochemical TEM cell was placed into the TEM, and upon in situ irradiation, no significant changes to the sample were observed (Supplementary Movie 1). The above result demonstrated that both alkaline solution and electron beam in TEM have little influence on the sample. The in situ liquid electrochemical TEM measurement was conducted under a constant current of -5 nA vs. Pt (Supplementary Fig. 25). After continuously subjecting the Ru-NiPS$_3$ NSs to a 2-h chronopotentiometry test, a significant reconstruction was observed at the edge position of the NSs (Fig. 4b, c). The corresponding SAED patterns clearly showed that while most of the NSs remained unchanged after the chronopotentiometry test (with similar diffraction spots as in Fig. 4d), a portion of the nanosheet underwent a transformation into a polycrystalline or amorphous state during the reconstruction process (as indicated by the presence of faint polycrystalline rings and amorphous halo ring in Fig. 4e). Detailed TEM images taken under different reaction time conditions show that the nanosheets underwent a gradual amorphization process, particularly at the edges (Supplementary Figs. 26 and 27). It is also found that the amorphization process would be more obvious at the thinner edges, which contribute to the formation of the functional amorphous layer for electrochemical reaction (Supplementary Fig. 28 and Supplementary Movie 2). According to some previous work, the amorphous layer can also protect the inner part of the nanosheet from over-etching, and effectively enhance the stability. Ex situ HRTEM images of Ru-NiPS$_3$ after stability tests for different reaction duration (from 1 to 16 h) are demonstrated in Supplementary Fig. 29, which showed that with the reaction time increases, the thickness of the in situ formed amorphous layer gradually increases, eventually reaching a roughly stable thickness (~7.5 nm for 16 h). According to some previous work, the amorphous layer can also protect the inner part of the nanosheet from over-etching, and effectively enhance the stability, which is proved by the in situ liquid TEM, which demonstrated that after the amorphization process is complete, the morphology and edges of the material undergo minimal observable changes (Supplementary Movie 3).

Additional characterizations of the morphology and electronic structure of Ru-NiPS$_3$ were conducted after the HER stability test, which were consistent with the in situ TEM characterizations. Compared with the pristine Ru-NiPS$_3$ NSs, the edges of the NSs were partially transformed into an amorphous structure after the HER stability test (Fig. 5a, b), while the inner part still maintains the crystalline structure (Fig. 5c, and Supplementary Fig. 30). According to the HAADF-STEM image (Fig. 5b), Ru atoms were mainly distributed at the edge of the NSs. Furthermore, HAADF-STEM EDS element mapping revealed the uniform distribution of Ni, P, and S, while Ru atoms were found primarily in the amorphous region, forming a Ru-enriched shell (Fig. 5d). PXRD was used to characterize the crystal structure of the Ru-NiPS$_3$ electrode after the HER stability test, which showed no significant change compared to the pristine sample. This finding suggests

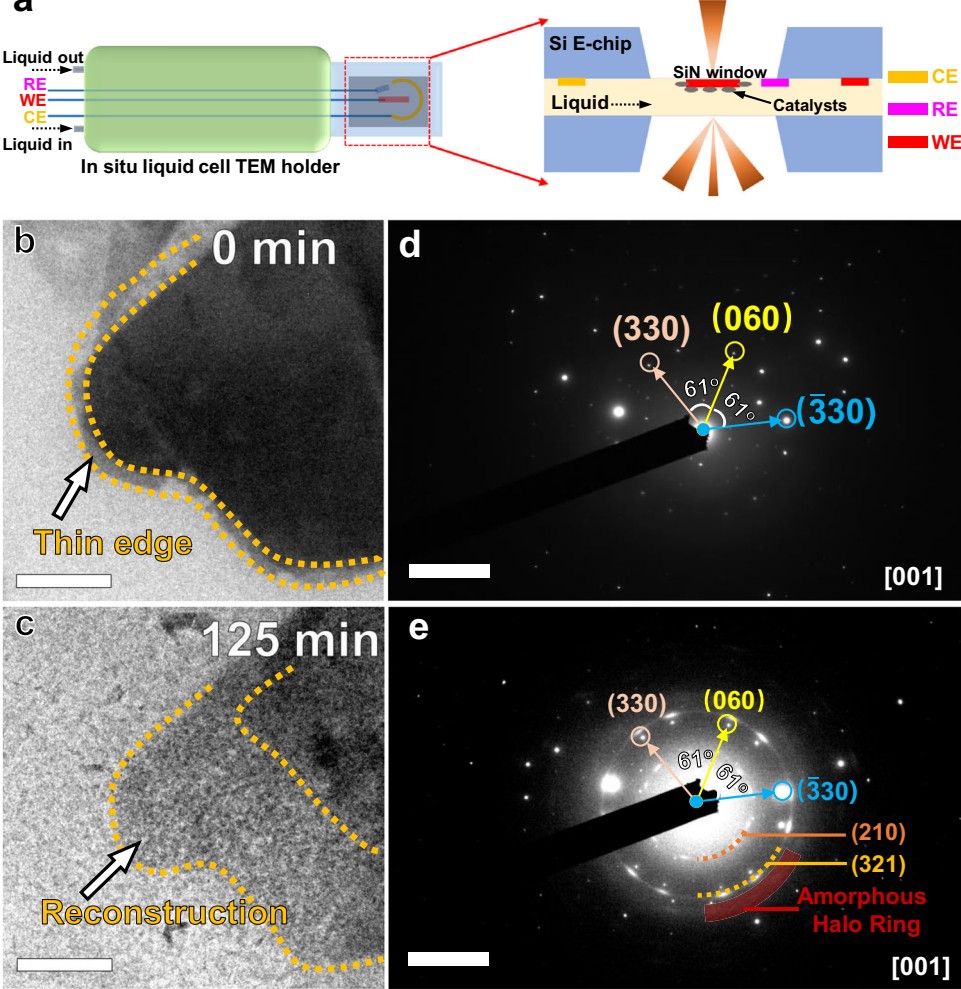

**Fig. 4 | The structural evolution of Ru-NiPS₃ NSs during the HER process.**
**a** Schematic illustration of the in situ electrochemical liquid cell TEM holder and the liquid cell. In situ liquid TEM image (scale bar: 0.2 μm) of Ru-NiPS₃ NSs (**b**) before and (**c**) after chronopotentiometry test. **d**, **e** demonstrated the corresponding SAED patterns (scale bar: 5 1/nm) for (**b**) and (**c**), respectively. The in situ TEM image provided clear evidence of a reconstruction process occurring at the edges of the NSs during the HER process. Additionally, the corresponding SAED pattern confirmed that a portion of the NSs underwent a transformation into a polycrystalline and amorphous state.

that the central part of the NSs remained as NiPS₃, which would contribute to the stability of the NSs (Fig. 5e).

To further study the surface chemical state of the Ru-NiPS₃ NSs after the HER stability test, Raman and XPS were employed. Raman spectra revealed only two broad peaks at approximately 347 cm⁻¹ and 490 cm⁻¹, which were attributed to Ni-S bonds and Ni oxides, respectively[35–37], indicating the surface reconstruction of NiPS₃ NSs (Fig. 5f). The Ni $2p_{3/2}$ spectra could be deconvoluted into three peaks. The peak located at 858.1 eV and 857.6 eV is attributed to the $Ni^{2+}$ in Ni-S species and Ni-O bond, respectively, which is consistent with the Raman result (Fig. 5g)[38,39]. Different from the pristine Ru-NiPS₃ NSs, only the peaks belonging to P−O bond were detected after HER test, which is inevitable during the alkaline HER process (Fig. 5h)[40,41]. The S $2p$ spectra were deconvoluted into two spin-orbit doublets, which correspond to the terminal $S^{2-}$ (-162.1 eV and 162.9 eV) and bridging $S_2^{2-}$ (-163.9 eV and 164.6 eV)[42–45]. Another broad peak at -169.6 eV was deconvoluted into two different peaks which belong to the sulfate species $[SO_4]^{2-}$ (at -170.8 eV) and sulfite species $[SO_3]^{2-}$ (at -168.5 eV) (Fig. 5i)[46]. In contrast to the pristine Ru-NiPS₃, the Ru $3p_{3/2}$ spectra after the HER stability test exhibited only a single broad peak at around 463.1 eV, which was assigned to the $Ru^{4+}$ species. Further ICP-OES measurements confirmed that the Ru species gradually dissolved into the electrolyte, and the remaining $Ru^{4+}$ species served as the active species for the HER process (Supplementary Fig. 31). This observation suggests that the $Ru^0$ or Ru cluster was unstable during the HER test, and only $Ru^{4+}$ species remained in the NiPS₃ NSs (Fig. 5j)[47,48].

To further support the crucial role of Ru-enriched edges, the structure of undoped NiPS₃ NSs was also characterized. Supplementary Figs. 32 and 33 demonstrate the presence of amorphous layer in the NiPS₃ NSs after the HER stability test. However, the HER activity of NiPS₃ was significantly lower than that of Ru-doped NSs, indicating the importance of Ru in the amorphous layer of Ru-NiPS₃ NSs.

## Understanding the role of surface amorphous layer

DFT calculations were conducted to elucidate the origin of the enhanced HER performance for Ru-NiPS₃ NSs. In alkaline media, the rate-determining step (RDS) of HER is typically determined by the adsorption energy of reaction intermediates[49,50]. According to Sabatier's principle, the optimal HER activity is achieved when the adsorption energy of the reaction intermediates is neither too high nor too low on the active sites[51,52]. Based on this perspective, we further investigated the influence of the coexistence of bridging $S_2^{2-}$ species and the Ru atoms in the amorphous layer on the adsorption energy of reactants. We first studied the undoped model with $S_2^{2-}$ group at the

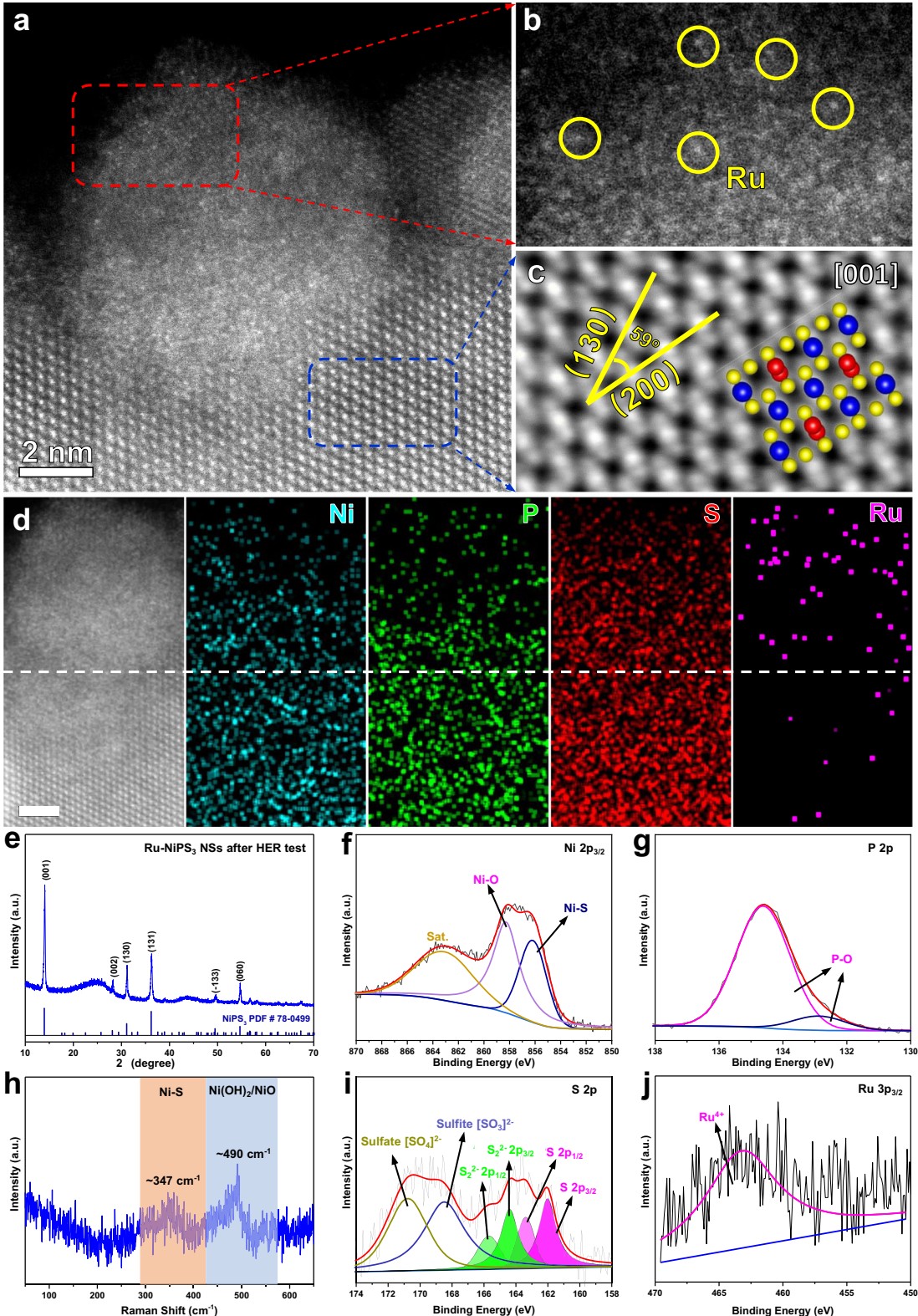

**Fig. 5 | Characterizations of Ru-NiPS₃ NSs after stability test. a** HAADF-STEM image of Ru-NiPS₃ after long-term stability HER test in 1 M KOH. **b, c** the enlarged part of amorphous part and inner part of Ru-NiPS₃ from (**a**). **d** EDS mapping of Ni, P, S, and Ru element (scale bar: 2 nm). **e** PXRD image of Ru-NiPS₃ NSs after HER test. **f** Raman spectrum of Ru-NiPS₃ NSs after HER test. **g–j** XPS spectra for Ni 2p$_{3/2}$, P 2p, S 2p, and Ru 3p$_{3/2}$ of Ru-NiPS₃ NSs after HER test.

edge sites (the model was shown in Supplementary Fig. 34). The calculation results revealed that the hydrogen adsorption energies at the eight potential active sites were neither too strong (<−0.50 eV) nor too weak (>0.67 eV), which are all unfavorable for the HER[13,53].

To explore the effects of Ru doping at the edge sites, three possible doping models with relatively stable structures were constructed and named NiPS₃-ac-Ru-1, 2, and 3 (Supplementary Table 5). According to the calculation results, Ru doping led to the optimization of

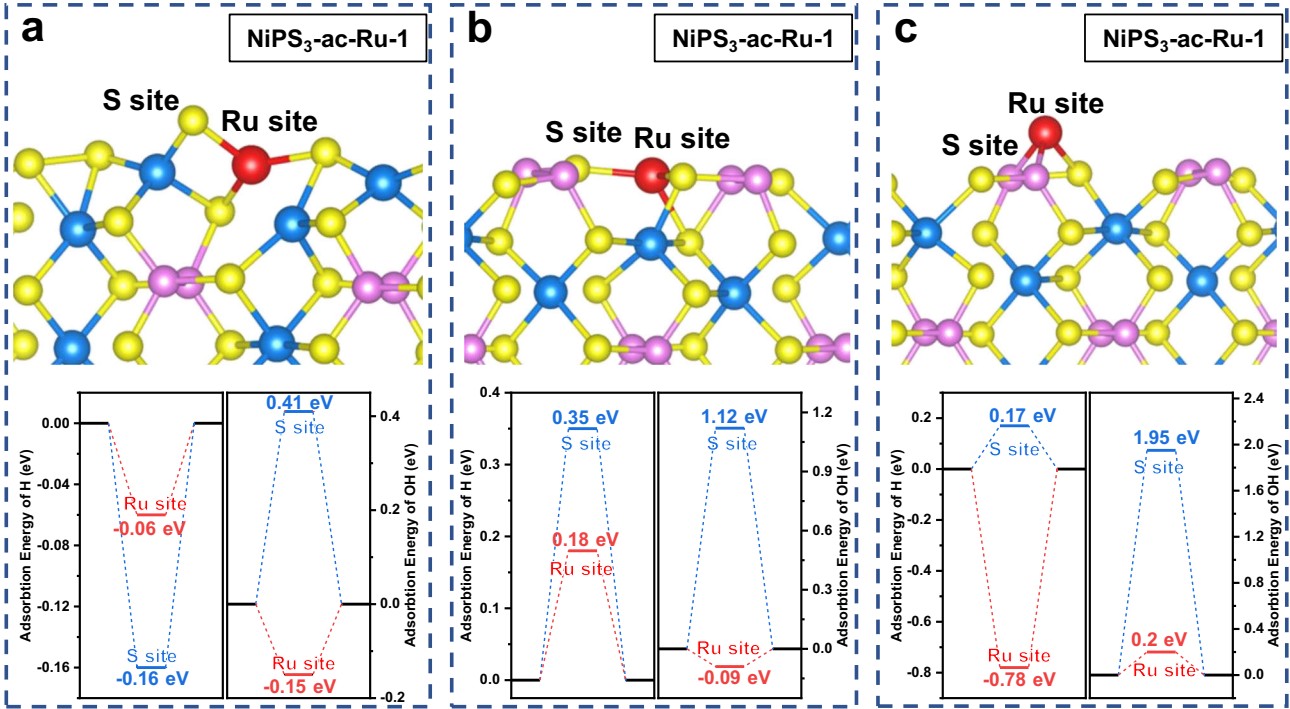

**Fig. 6 | DFT calculation results of H* and OH* adsorption energy at Ru and adjacent S sites with typical configurations. a** Ru atom is doped into the bridging $S_2^{2-}$ layer. **b** Ru atom is doped into other edge sites without bridging $S_2^{2-}$. **c** Ru atom is adsorbed onto the edge of NiPS$_3$.

hydrogen adsorption energies to a favorable range (between −0.16 eV and 0.35 eV), which was superior to the undoped models (Fig. 6a, b). Nonetheless, if the Ru atom is simply adsorbed on the edges of NiPS$_3$, the H* adsorption ability would remain too strong (about −0.78 eV), impeding the desorption of the H* atom for the subsequent step (Fig. 6c). The adsorption ability of OH* species is another crucial factor in evaluating the HER performance in alkaline electrolytes, as it affects the dissociation process of H$_2$O molecules[5]. The ability of OH* to adsorb onto the active sites of the catalyst can impact the overall reaction rate and efficiency. Therefore, on the basis of appropriate H* adsorption energy, it is necessary to further consider the adsorption ability of active sites for OH*[54]. The DFT calculation results demonstrate that if the Ru atom was doped into the $S_2^{2-}$ layer (NiPS3-ac-Ru-1), the dissociated H* and OH* species would be adsorbed on S site (−0.16 eV) and adjacent Ru site (−0.15 eV) respectively. However, in the NiPS$_3$-ac-Ru-2 and NiPS$_3$-ac-Ru-3 models, both H* and OH* were found to preferentially adsorb onto the Ru sites, leading to competition for adsorption between the two species and potentially resulting in the poisoning of active sites[55]. Moreover, NiPS$_3$-ac-Ru-1 demonstrated stronger adsorption ability for the Ru atom, resulting in a more stable structure (Supplementary Table 5). The DFT calculation results further proved that the formation of the $S_2^{2-}$ the group could effectively stabilize the doped Ru atom. Additionally, the separation of adsorption sites for H* and OH* can help prevent the poisoning of active sites while moderating the adsorption ability of reaction intermediates. These combined effects ultimately enhance the catalytic HER activity of Ru-NiPS$_3$ NSs in alkaline electrolytes.

In summary, this study has demonstrated the significant role of the in situ formed amorphous layer and proposed an edge site decoration strategy to enhance the alkaline HER performance in layered catalysts. The in situ liquid electrochemical TEM was used to directly observe the amorphization in Ru-NiPS$_3$, which is often overlooked in investigations of the origin of catalytic activity for metal phosphorus trichalcogenides. Theoretical calculations and experimental characterizations indicated that the formation of a surface amorphous layer with abundant bridging $S_2^{2-}$ species can stabilize the doped Ru

atoms and reduce the leaching of functional groups during the HER process in alkaline electrolytes. Moreover, the doped Ru species located at the amorphous layer can modulate the adsorption energy of reaction intermediates and accelerate the water dissociation process, ultimately enhancing the alkaline HER activity. This work provides insights into the function of reconstructed amorphous layers in transition metal-based electrocatalysts and proposes a proper modulation strategy to improve the catalytic activity of layered materials.

## Methods
### Chemicals
All chemicals used in this study were of analytical grade and purchased from Aladdin Biochemical Technology Co., Ltd. in Shanghai, China. No further purification was carried out on any of the chemicals prior to use. Carbon cloth (CC) was obtained from Shanghai Hesen Electric Co., Ltd. and used in its as-received form.

### Synthesis of NiPS$_3$ and Ru-NiPS$_3$ electrodes
In this experiment, Ni precursors were grown on carbon cloth (CC) using a hydrothermal method[56]. Typically, 2 mmol Ni(NO$_3$)$_2$·6H$_2$O, 10 mmol urea, and 5 mmol NH$_4$F were dissolved in 40 mL of deionized (DI) water. After continuously stirring for 30 min, the solution was transferred into a 50 mL Teflon-lined stainless-steel autoclave. A piece of CC (with an area of 2 cm × 3 cm) was thoroughly cleaned with acetone, water, and ethanol to remove the possible contaminant on the surface. Then the CC was also transferred into the autoclave. The sealed autoclave was sealed and heated at 130 °C for 8 h, and then the CC was taken out. Afterward, the CC was cleaned with deionized water (DI) water and ethanol, and the precursor was dried in a vacuum at 60 °C for 8 h. To synthesize the NiPS$_3$ NSs, 40 mg mixture of red phosphorus and sulfur powder, with a stoichiometric ratio of 1:3, was vacuum sealed in a quartz tube, together with the Ni precursor. Then the quartz tube was firstly heated at 300 °C for 30 min and then heated at 450 °C for 5 h, with a heating rate of 5 °C/min. After cooling to room temperature, the sample was washed with DI water and ethanol and dried in vacuum at 60 °C for further characterization. To synthesize

the Ru-NiPS$_3$ NSs, the Ni precursor was first immersed into the RuCl$_3$ aqueous solution (3 mg ml$^{-1}$) for different durations to finish the ion-exchange process. The resulting Ru-Ni precursor was then heated using a similar procedure to the synthesis of NiPS$_3$ NSs. Since the doping amount of Ru is relatively low, the final loading of electrocatalysts on the electrode is about 1.5 mg cm$^{-2}$. For comparison, NiPS$_3$ single crystal was also fabricated with typical CVT methods following the description of previous work[57].

## Materials characterizations

The crystalline phase and compositions were characterized by XRD (Rigaku SmartLab 9 kW) equipped with a $\lambda = 1.54056$ Å Cu Kα1 radiation source. Field emission scanning electron microscope (Thermo Fisher Scientific™ Helios 5 CX DualBeam™ Scanning Electron Microscope) was utilized to characterize the morphologies of the precursor and the final (Ru-)NiPS$_3$ NSs. XPS spectra were collected on a Thermo Fisher Scientific Nexsa. The transmission electron microscope (JEOL, JEM-2100F; acceleration voltage, 200 kV), and double spherical aberration-corrected transmission electron microscope (Thermo Fisher Spectra 300, operated at 300 kV) were employed to investigate the morphologies, microstructures, and elements distribution. Dr. Probe was used for simulating STEM-HAADF images. Accelerating voltage, convergence semi-angle, and collection angle were set the same as the imaging, which were 300 kV, 15 mrad, and 35–200 mrad, respectively. Raman spectroscopy was conducted on a WITec, alpha300 R equipment (with a 532 nm laser). EPR spectra were obtained on a Bruker, A300-10-12 spectrometer. Atomic force microscopy (AFM) measurement was conducted on an AFM5300E system (HITACHI, Japan).

## In situ liquid electrochemical TEM characterization

To observe the in situ morphological evolution of Ru-NiPS$_3$ NSs, a JEM-2100F electron microscope operated at 200 kV was used in conjunction with a liquid TEM holder (Protochips, Poseidon Select). The electron flux was calculated to be about 20 e$^-$ Å$^{-2}$ s$^{-1}$. The liquid electrochemical chips (Protochips E-chips, ECT-45CR) are composed of two silicon chips, which are washed in acetone, methanol, and ethanol for 5 min, respectively, to remove the protective coating. Ru-NiPS$_3$ ink was then dropped onto the silicon nitride (SiN$_x$) window, and the chips were further cleaned with Ar/O$_2$ plasma for 30 s. The in situ observation was conducted in an alkaline media solution (0.1 M NaOH), and a Gamry 600+ potentiostat (Gamry, Warminster, PA) was used to provide a constant current of -5 nA vs. Pt during the whole observation. The electrolyte flow rate is controlled at 200 μL h$^{-1}$ to avoid damaging the SiN$_x$ window. In order to obtain clear TEM images with good spatial resolution, the samples in these in situ liquid TEM experiments were imaged within a thin liquid layer. It should be noted that the applied experimental conditions in these studies may not perfectly replicate those of a realistic electrochemical cell. As a result, there may be minor differences between the results obtained from in situ and ex situ measurements.

## Electrochemical measurements

All electrochemical measurements were conducted using a typical three-electrode cell with a CHI 760E electrochemical workstation (CH Instruments, Inc. Shanghai). The as-systemized electrode, Hg/HgO electrode, and graphite rod (Alfa Aesar, 99.9995%) were used as the working electrode, reference electrode, and counter electrode, respectively. Before the LSV test in 1 M KOH electrolyte, all electrodes were activated by the cyclic voltammetry (CV) technique for 100 cycles to obtain stable LSV curves. The scan rate is 50 mV s$^{-1}$, within the potential range from −0.8 V vs. Hg/HgO to −1.5 V vs. Hg/HgO, and the total activation time is about 1 h. To avoid the influence of the Ru species dissolved in the electrolyte during the activation process, the electrode was replaced immediately when the activation process is finished. LSV curves were then obtained with a scan rate of 2 mV s$^{-1}$. In this work, all potentials were converted to RHE with the equation $E_{RHE} = E_{Hg/HgO} + 0.098\,V + 0.059 \times pH$. EIS measurements were carried out within a frequency range of $10^6$ Hz to $10^{-2}$ Hz, and the charge transfer resistance ($R_{ct}$) obtained by fitting the EIS data was used for the $iR$ correction.

## Calculation details

To investigate the geometries and electronic properties of Ru-doped NiPS$_3$ materials, spin-polarized density functional theory (DFT) calculations were conducted using the Vienna ab initio Simulation Package (VASP) program package[58,59] with the projector augmented wave (PAW)[60]. The exchange-correlation interactions were described using the Perdew, Burke, and Ernzernhof (PBE) functional[60] with the generalized gradient approximation (GGA)[61]. The kinetic energy cutoff for the plane-wave basis set was set to 400 eV, and the distance of the vacuum layer was greater than 20 Å to prevent interlayer interactions. To correct for van der Waals interactions on the surface, the DFT-D3 scheme of Grimme was applied[62]. The electronic SCF tolerance was set to $10^{-4}$ eV. Fully relaxed geometries and lattice constants were obtained by optimizing all atomic positions until the Hellmann–Feynman forces were less than 0.04 eV/Å. The structural optimizations used a gamma-centered Monkhorst–Pack scheme with k-point samplings of $2 \times 1 \times 1$[61]. To convert the calculated DFT adsorption energies ($\Delta E$) into Gibbs free energies ($\Delta G$) for H (0.24 eV)[63] and OH* (0.29 eV)[64], entropic (TS) and zero-point energy (ZPE) corrections were applied to the adsorbed species.

## Data availability

The data that support the findings of this study are available from the corresponding author upon reasonable request.

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

## Acknowledgements

This work was supported by the National Science Foundation of China (Project Nos. 52173230, 52222218, 52272045, 21703076), The Hong Kong Research Grant Council General Research Fund (Project Nos. 11312022, 15302522, 11300820), Environment and Conservation Fund (Project Nos. 69/2021, 34/2022), City University of Hong Kong (Project Nos. 6000758, 9211308, 9667223, 9678303), The State Key Laboratory of Marine Pollution (SKLMP) Seed Collaborative Research Fund SKLMP/SCRF/0037, The Hong Kong Polytechnic University (Project No. ZVH0, SAC9), the Research Institute for Advanced Manufacturing of The Hong Kong Polytechnic University, Shenzhen Science, Technology and Innovation Commission (Project No. JCYJ20200109110213442), Natural Science Foundation of Jiangsu Province of China (Project No. BK20211609).

## Author contributions

J.Z., T.H.L. designed and supervised the research project. Q.F. carried out the synthesis with assistance from W.Q.S. and C.S.T. Q.F., and W.Q.S. carried out the electrochemical measurements. Q. F., F. Y. Z., and K.H.L conducted the in situ TEM characterizations. Q.F., L.K.W., and X.D.Z. analyzed the TEM data. Q.M.D. carried out the DFT calculations and analysis. J.Z. and Q.F. wrote and revised the manuscript.

## Competing interests

The authors declare no competing interests.
