## [Peer Review File · Nature Communications]

Unraveling and Leveraging in situ Surface Amorphization for Enhanced Hydrogen Evolution Reaction in Alkaline MediaREVIEWER COMMENTS

Reviewer #1 (Remarks to the Author):

In the submitted paper, Fu et al. presented results regarding the synthesis of Ru-NiPS3 nanosheets, their characterization and electrochemical properties toward HER. They also proposed that the property improvement observed can be attributed to amorphization of the nanosheets. Here, I will mainly comment on the electron microscopy data (both ex situ and in situ) and their interpretations, which are both highly problematic.

1. The substitution of Ru into the lattice cannot be directly elucidated from the intensity profiles as the authors did in Figure 1f and Supplementary Figure 11. The samples are too thick for such a simple interpretation. Here, there are two common ways to infer the presence of heavier atoms.

a. Look at the samples that are not under strong channeling conditions (i.e. absence of any atomic resolution in the support). However, it also means that there is no way to determine if the Ru simply sits on the surface or substitutes for Ni

b. The other way is to perform image simulations for comparison to show that the intensity difference is indeed consistent with the substitution of a Ni atom by a Ru atom.

2. The liquid cell experiments are interpreted wrongly and have significant issues.

a. The electron flux used for imaging is a critical experimental parameter that needs to be declared in every paper using the technique, which is currently absent in the paper.

b. The experiments also appeared to be performed under thin liquid conditions, which is not ideal for electrochemistry experiments. The authors should at least mention that they did not work with a fully filled cell.

c. The amorphization cannot be inferred from the electron diffraction patterns. It is clear from Supplementary Movie 1, the NS is rotating/rolling during the experiment (also from the fading in and out of diffraction contrast). The loss of intensity is due to the NS moving out of strong diffraction conditions and not because the sample became amorphous.

d. The center diffraction spot in 4(d) is also highly stigmatic (i.e. not circular), the diffraction rings acquired under such conditions will be elliptical and not suitable for reliable indexing.

e. It is also common practice to use a beam blocker to blank the central beam. The main reason (apart from avoiding damage to the camera) is that with an intense central beam takes up most of the dynamic range of the detector, which makes it very difficult to pick up weaker diffraction spots.

3. Hence, the entire discussion on page 10 regarding the difference between the behavior on Pt and on GC is (likely) wrong. The more plausible hypothesis is that the formation of hydrogen bubbles on the Pt pushes the NS and causes it to move or curl whereas the sample on the GC is stagnant.

4. Did the authors acquire DPs from the sample on GC?

5. The author's claim of amorphization based on liquid cell data is, in fact, not self-consistent with their own ex situ analysis.

a. The AC-STEM shows that most of the NS remains crystalline and any amorphization is superficial.

b. This is also supported by the negligible difference powder diffraction patterns.

Other comments

6. The authors should mention in the main text the average thickness of the NSs.

7. Line 110. The authors specify that that the d-spacings for are the (130) and (1-30) of NiPS3.

8. Why are the chronopotentiometry experiments (@100 mA and @5 nA) fixed at positive current densities? Shouldn't the currents?

9. Authors should also provide the chronopotentiometry data from the liquid cell holder.

Reviewer #2 (Remarks to the Author):

In this manuscript, the authors synthesized a Ru-NiPS3 nanosheets (Ru-NiPS3 NSs) catalyst by three-step procedure for the hydrogen evolution reaction (HER). The samples were characterized by SEM, EDS, XPS, XRD and in situ TEM. The catalyst properties have been systematically discussed through

electrochemical test and DFT, which well-demonstrates the importance of the highly active amorphous surfaces. Thus, publication this work on nature communications could be recommended after carefully addressing the following issues regarding material characterizations and proposed mechanism.

1. Could the surface amorphization of the catalyst be controlled accurately through the method provided in this paper? The detailed experimental process should be provided for others to repeat.
2. Does the samples immerse in RuCl₃ solution for different durations affect the loading amount of Ru in the nanosheets. Is the Ru-NiPS₃ with 0.81wt% content of Ru the best one after optimization? The related information should be provided.
3. Why are the Ru atoms mainly distributed at the edge of the NSs, forming a Ru-enriched shell? What is the mechanism in forming such kind of structure?
4. The authors demonstrated that the Ru₀ or Ru cluster was unstable during the HER test, and only Ru₄₊ species remained at the edge sites of NiPS₃ NSs (Fig. 5j). So, are these Ru₀ or Ru cluster species converted to Ru₄₊ or dissolved in the electrolyte? Further experimental verification is suggested.
5. Is all Ru scattered around the edges after electrolysis? Does the amorphous state form only on the [001] crystal plane (Fig. 5a)? Or are there amorphous states in other crystal planes? It is recommended to supplement other regional HAADF images.
6. From Fig. 4 it is hesitant to conclude anything from the in situ electrochemical liquid cell TEM holder. The TEM images and the contrast are not convincing enough.
7. In the XPS, the same element fitting should be redone with all components having the same FWHM. In Figure 5 should be refitted it.

Reviewer #4 (Remarks to the Author):

In this work, the authors aim to address a commonly overlooked issue in the alkaline HER: the formation of an amorphous layer during the reaction process. The lack of evidence for the in-situ characterization of the amorphous layer formed during the reaction has hindered the full development of its essential role and potential applications. Fu et al. have utilized an advanced in-situ liquid TEM technique to directly demonstrate the formation of the amorphous layer, which provided direct evidence of the amorphization. Their results prove that the amorphous layer is significant in analyzing the true catalytic mechanism and active sites for the HER process. Furthermore, the authors have also shown, both theoretically and experimentally, that by rationally designing and utilizing the unavoidable amorphous layer, the catalytic performance of the electrocatalyst can be significantly improved. I highly recommend this manuscript for publication in Nature Communications. Please find below some detailed comments for the authors to consider.

1. The manuscript was written mainly to discuss the amorphous layer, will the sample be oxidized during the storage? And if the sample was oxidized after the preparation, will the oxidized layer have any influence on the catalytic process?
2. As shown in Figure 2g, some Ru(0) was detected before the HER test. I think this should belong to the Ru cluster or Ru metal. Will the Ru(0) have any influence on the catalytic performance?
3. The in-situ TEM images and movies have demonstrated that the NiPS₃ NSs undergo an amorphization process. Is it possible that the electrolyte could have an impact on the formation of the amorphous layer on the catalyst surface?
4. In Figure 1e, the author used a CIF profile to demonstrate the crystal structure of NiPS₃. It would be better if changed it into a HAADF-STEM simulation image.
5. Some typos in the manuscript should be corrected. For instance, in line 57, the word "lead" should be "leads". In line 115 "characterization is conducted" should be "characterization was conducted". In line 355, it is recommended to spell out the abbreviation "DI" in its complete form, when it is used for the first time in a document.

REVIEWER COMMENTS

Reviewer #1 (Remarks to the Author):

In the submitted paper, Fu et al. presented results regarding the synthesis of Ru-NiPS3 nanosheets, their characterization and electrochemical properties toward HER. They also proposed that the property improvement observed can be attributed to amorphization of the nanosheets. Here, I will mainly comment on the electron microscopy data (both ex situ and in situ) and their interpretations, which are both highly problematic.

1. The substitution of Ru into the lattice cannot be directed elucidated from the intensity profiles as the authors did in Figure 1f and Supplementary Figure 11. The samples are too thick for such a simple interpretation. Here, there are two common ways to infer the presence of heavier atoms.

a. Look at the samples that are not under strong channeling conditions (i.e. absence of any atomic resolution in the support). However, it also means that there is no way to determine if the Ru simply sits on the surface or substitutes for Ni

b. The other way is to perform image simulations for comparison to show that the intensity difference is indeed consistent with the substitution of a Ni atom by a Ru atom.

2. The liquid cell experiments are interpreted wrongly and have significant issues.

a. The electron flux used for imaging is a critical experimental that needs to be declared in every paper using the technique, which is currently absent in the paper.

b. The experiments also appeared to be performed under thin liquid conditions, which is not ideal for electrochemistry experiments. The authors should at least mention that they did not work with a fully filled cell.

c. The amorphization cannot be inferred from the electron diffraction patterns. It is clear from Supplementary Movie 1, the NS is rotating/rolling during the experiment (also from the fading in and out of diffraction contrast). The loss of intensity is due to the NS moving out of strong diffraction conditions and not because the sample became amorphous.

d. The center diffraction spot in 4(d) is also highly stigmatic (i.e. not circular), the diffraction rings acquired under such conditions will be elliptical and not suitable for reliable indexing.

e. It is also common practice to use a beam blocker to blank the central beam. The main reason (apart from avoiding damage to the camera) is that with an intense central beam takes up most of the dynamic range of the detector, which makes it very difficult to pick up weaker diffraction spots.

3. Hence, the entire discussion on page 10 regarding the difference between the behavior on Pt and on GC is (likely) wrong. The more plausible hypothesis is that the formation of hydrogen bubbles on the Pt pushes the NS and causes it to move or curl whereas the sample on the GC is stagnant.

4. Did the authors acquire DPs from the sample on GC?

5. The author's claim of amorphization based on liquid cell data is, in fact, not self-consistent with their own ex situ analysis.

a. The AC-STEM shows that most of the NS remains crystalline and any amorphization is superficial.

b. This is also supported by the negligible difference powder diffraction patterns.

Other comments

6. The authors should mention in the main text the average thickness of the NSs.

7. Line 110. The authors specify that that the d-spacings for are the (130) and (1-30) of NiPS₃.

8. Why are the chronopotentiometry experiments (@100 mA and @5 nA) fixed at positive current densities? Shouldn't the currents?

9. Authors should also provide the chronopotentiometry data from the liquid cell holder.

Reviewer #2 (Remarks to the Author):

In this manuscript, the authors synthesized a Ru-NiPS₃ nanosheets (Ru-NiPS₃ NSs) catalyst by three-step procedure for the hydrogen evolution reaction (HER). The samples were characterized by SEM, EDS, XPS, XRD and in situ TEM. The catalyst properties have been systematically discussed through electrochemical test and DFT, which well-demonstrates the importance of the highly active amorphous surfaces. Thus, publication this work on nature communications could be recommended after carefully addressing the following issues regarding material characterizations and proposed mechanism.

1. Could the surface amorphization of the catalyst be controlled accurately through the method provided in this paper? The detailed experimental process should be provided for others to repeat.
2. Does the samples immerse in RuCl₃ solution for different durations affect the loading amount of Ru in the nanosheets. Is the Ru-NiPS₃ with 0.81wt% content of Ru the best one after optimization? The related information should be provided.
3. Why are the Ru atoms mainly distributed at the edge of the NSs, forming a Ru-enriched shell? What is the mechanism in forming such kind of structure?
4. The authors demonstrated that the Ru⁰ or Ru cluster was unstable during the HER test, and only Ru⁴⁺ species remained at the edge sites of NiPS₃ NSs (Fig. 5j). So, are these Ru⁰ or Ru cluster species converted to Ru⁴⁺ or dissolved in the electrolyte? Further experimental verification is suggested.
5. Is all Ru scattered around the edges after electrolysis? Does the amorphous state form only on the [001] crystal plane (Fig. 5a)? Or are there amorphous states in other crystal planes? It is recommended to supplement other regional HAADF images.
6. From Fig. 4 it is hesitant to conclude anything from the in situ electrochemical liquid cell TEM holder. The TEM images and the contrast are not convincing enough.
7. In the XPS, the same element fitting should be redone with all components having the same FWHM. In Figure 5 should be refitted it.

Reviewer #4 (Remarks to the Author):

In this work, the authors aim to address a commonly overlooked issue in the alkaline HER: the formation of an amorphous layer during the reaction process. The lack of evidence for the in-situ characterization of the amorphous layer formed during the reaction has hindered the full development of its essential role and potential applications. Fu et al. have utilized an advanced in-situ liquid TEM technique to directly demonstrate the formation of the amorphous layer, which provided direct evidence of the amorphization. Their results prove that the amorphous layer is significant in analyzing the true catalytic mechanism and active sites for the HER process. Furthermore, the authors have also shown, both theoretically and experimentally, that by rationally designing and utilizing the unavoidable amorphous layer, the catalytic performance of the electrocatalyst can be significantly improved. I highly recommend this manuscript for publication in Nature Communications. Please find below some detailed comments for the authors to consider.

1. The manuscript was written mainly to discuss the amorphous layer, will the sample be oxidized during the storage? And if the sample was oxidized after the preparation, will the oxidized layer have any influence on the catalytic process?
2. As shown in Figure 2g, some Ru(0) was detected before the HER test. I think this should belong to the Ru cluster or Ru metal. Will the Ru(0) have any influence on the catalytic performance?
3. The in-situ TEM images and movies have demonstrated that the NiPS3 NSs undergo an amorphization process. Is it possible that the electrolyte could have an impact on the formation of the amorphous layer on the catalyst surface?
4. In Figure 1e, the author used a CIF profile to demonstrate the crystal structure of NiPS3. It would be better if changed it into a HAADF-STEM simulation image.
5. Some typos in the manuscript should be corrected. For instance, in line 57, the word “lead” should be “leads”. In line 115” characterization is conducted” should be “characterization was conducted”. In line 355, it is recommended to spell out the abbreviation "DI" in its complete form, when it is used for the first time in a document.

RESPONSE TO REVIEWER COMMENTS

Title: “Unraveling and Leveraging *in situ* Surface Amorphization for Enhanced Hydrogen Evolution Reaction in Alkaline Media”

(Manuscript ID: NCOMMS-23-21960)

We sincerely thank all reviewers for their valuable comments and suggestions, which are certainly helpful for improving the quality of our manuscript. We have thoroughly reviewed and addressed all the points raised by the reviewers in a systematic and careful manner. A point-by-point response to all comments is given below. For clarity, the reviewer’s comments are in black text, and our responses are in blue text. Additions and revisions to the manuscript and SI have been included in this response, and they are given in red text.

Reviewer #1 (Remarks to the Author):

In the submitted paper, Fu et al. presented results regarding the synthesis of Ru-NiPS3 nanosheets, their characterization and electrochemical properties toward HER. They also proposed that the property improvement observed can be attributed to amorphization of the nanosheets. Here, I will mainly comment on the electron microscopy data (both *ex situ* and *in situ*) and their interpretations, which are both highly problematic.

Response: We are grateful to the reviewer for the thorough review of our manuscript, and we have carefully considered the feedback regarding the *in-situ* TEM section. As per the suggestions, we have conducted additional experiments and made significant revisions to this section, which are outlined below.

1. The substitution of Ru into the lattice cannot be directed elucidated from the intensity profiles as the authors did in Fig. 1f and Supplementary Fig. 11. The samples are too thick for such a simple interpretation. Here, there are two common ways to infer the presence of heavier atoms.

a. Look at the samples that are not under strong channeling conditions (i.e. absence of any atomic resolution in the support). However, it also means that there is no way to determine if the Ru simply sits on the surface or substitutes for Ni

b. The other way is to perform image simulations for comparison to show that the intensity difference is indeed consistent with the substitution of a Ni atom by a Ru atom.

Response: Thank you for your suggestion. We have incorporated your feedback into the revised manuscript by rearranging the TEM image in Fig. 1. We have added simulated HAADF-STEM image to Fig. 1 as Fig. 1g (the EDS mapping thus changed in Fig. 1i). The modified figure is demonstrated as Fig. R1-1.

Fig. R1-1 (a) SEM image of the as-prepared Ru-NiPS₃ NSs. (b), (c) HRTEM image and the corresponding SAED image along the [001] zone axis. (d) Atomic-level HAADF-STEM image of an ultrathin NiPS₃ nanosheet. (e) Crystal structure of NiPS₃ along [001] zone axis. (f) Line intensity profile obtained from the selected area in (d). (g) and (h) are the simulated HAADF-STEM image with Ru doped into the NiPS₃ lattice and the corresponding line intensity profile which is similar to the experimental one. (i) HEDF-STEM image corresponding EDS mapping of Ru-NiPS₃ nanosheet (scale bar 500 nm).

In the manuscript, we added the description of the simulation: Line 118 “.....which is consistent with the simulated HAADF image and the corresponding intensity profile

(Fig. 1g, and h)”

The simulation detail with Dr. Probe software package is also added: Line 389 *“Dr. Probe was used for simulating STEM-HAADF images. Accelerating voltage, convergence semi-angle, and collection angle were set same as the imaging, which were 300 kV, 15 mrad, and 35-200 mrad, respectively.”*

2. The liquid cell experiments are interpreted wrongly and have significant issues.

a. The electron flux used for imaging is a critical experimental that needs to be declared in every paper using the technique, which is currently absent in the paper.

b. The experiments also appeared to be performed under thin liquid conditions, which is not ideal for electrochemistry experiments. The authors should at least mention that they did not work with a fully filled cell.

Response for a and b: We sincerely thank the reviewer for bringing to our attention the omission of our experiments. We agree with your perspective that the thin liquid conditions used in our study may differ from the conditions of a realistic electrochemical cell. However, the use of a "thin liquid strategy" is a commonly applied technique to obtain high-quality TEM images in liquid cells, as demonstrated in previous works (e.g., *ACS Nano* 2021, 15, 6, 10228–10240, *J. Am. Chem. Soc.* 2022, 144, 34, 15698–15708, *Nature* 614, 262–269 (2023)). Therefore, we followed this established approach to perform our in-situ liquid TEM measurements.

In the revised manuscript, we have emphasized this point in the Method section and provided more experimental information. We also provided additional experimental results of the in-situ TEM measurements. The experimental section has been revised and is presented below:

In situ liquid electrochemical TEM characterization

To observe the in situ morphological evolution of Ru-NiPS₃ NSs, a JEM-2100F electron microscope operated at 200 kV was used in conjunction with a liquid TEM holder (Protochips, Poseidon Select). The electron flux was calculated to be about $20 e^- \text{Å}^{-2} s^{-1}$. The liquid electrochemical chips (Protochips E-chips, ECT-45CR) are composed of two silicon chips, which are washed in acetone, methanol, and ethanol for

5 min, respectively, to remove the protective coating. Ru-NiPS₃ ink was then dropped onto the silicon nitride (SiN_x) window, and the chips were further cleaned with Ar/O₂ plasma for 30 seconds. The *in situ* observation was conducted in an alkaline media solution (0.1 M NaOH), and a Gamry 600+ potentiostat (Gamry, Warminster, PA) was used to provide a constant current of -5 nA vs. Pt during the whole observation. The electrolyte flow rate is controlled at 200 μL h⁻¹ to avoid damaging the SiN_x window. What should be mentioned is that, we adopted a currently commonly used "thin liquid strategy" to minimize the influence of liquid on the resolution of TEM, so that we can acquire some sufficiently clear TEM images during our experiments.

c. The amorphization cannot be inferred from the electron diffraction patterns. It is clear from Supplementary Movie 1, the NS is rotating/rolling during the experiment (also from the fading in and out of diffraction contrast). The loss of intensity is due to the NS moving out of strong diffraction conditions and not because the sample became amorphous.

d. The center diffraction spot in 4(d) is also highly stigmatic (i.e. not circular), the diffraction rings acquired under such conditions will be elliptical and not suitable for reliable indexing.

e. It is also common practice to use a beam blocker to blank the central beam. The main reason (apart from avoiding damage to the camera) is that with an intense central beam takes up most of the dynamic range of the detector, which makes it very difficult to pick up weaker diffraction spots.

Response for c, d and e: Thank you for your insightful comment. Regarding the *in situ* liquid TEM cell, we would like to acknowledge that the distribution of the sample on the electrodes is randomized, and the selection of the investigated sample is sometimes restricted. In our previous attempt at *in situ* liquid TEM measurement, we chose a relatively thin sample on the Pt electrode, as we speculated that the thinner sample would react more easily and provide a more intuitive understanding of the amorphization process. However, we agree with the reviewer's suggestion that there may be two shortcomings with this approach. Firstly, the Pt electrode is highly active

for HER, and the generated H₂ bubble around the Pt electrode may render the nanosheets unstable and provide inaccurate information. Secondly, the movement of the nanosheet makes it difficult to obtain an ideal SAED pattern.

Therefore, in the revised version, we conducted new round of *in-situ* liquid TEM measurement and have optimized the experimental conditions:

To minimize the potential impact of sample movement on our TEM imaging, we opted to use a relatively thick sample. Additionally, we carefully selected a sample located at the GC electrode, which we believe will allow us to obtain higher-quality TEM images. The final results are demonstrated in Fig. R1-2, and we replace the Fig. 4 in the revised manuscript with the modified one.

Fig. R1-2. (a) Schematic illustration of the *in situ* electrochemical liquid cell TEM holder and the liquid cell. *In situ* liquid TEM image (scale bar: 0.2 mm) of Ru-NiPS₃ NSs (b) before and (c) after chronopotentiometry test. (d) and (e) demonstrated the corresponding SAED patterns (scale bar: 5 1/nm) for (b) and (c), respectively.

3. Hence, the entire discussion on page 10 regarding the difference between the behavior on Pt and on GC is (likely) wrong. The more plausible hypothesis is that the formation of hydrogen bubbles on the Pt pushes the NS and causes it to move or curl whereas the sample on the GC is stagnant.

Response: We thank the reviewer for pointing out this issue. We have revised the discussion on the *in situ* electrochemical liquid cell TEM results based on the feedback from the reviewer. The modified discussion takes into account our new *in situ* liquid TEM measurement results, which are presented below, and the corresponding Fig. 4 has already presented as Fig. R1-2 in **Comment 2**:

Observation and characterization of surface reconstruction process. In situ liquid electrochemical TEM technique was conducted to gain insights into the structural evolution of Ru-NiPS₃ NSs during the alkaline HER process (Fig. 4a demonstrated the structure of the in situ electrochemical liquid cell TEM holder and the liquid cell). To minimize interference from other factors in the analysis, we first ruled out two potential sources of disturbance: corrosion of the sample by the alkaline solution, and interference from the electron beam on the sample. The Ru-NiPS₃ NSs electrode was first immersed into 1 M KOH for 24 h to show the influence of the alkaline electrolyte. As demonstrated in Supplementary Fig. 23a, the morphology demonstrated negligible variation after the immersing process, which is also proved by the corresponding XRD pattern (Supplementary Fig. 23b) and Raman spectra (Supplementary Fig. 23c). To rule out the influence of the electron beam in the TEM, the assembled liquid in situ electrochemical TEM cell was placed into the TEM, and upon in situ irradiation, no significant changes to the sample were observed (Supplementary Movie S1). The above result demonstrated that both alkaline solution and electron beam in TEM have little

influence on the sample. The in situ liquid electrochemical TEM measurement was conducted under a constant current of -5 nA vs. Pt (Supplementary Fig. 24). The in situ TEM images before and after 2 h continually chronopotentiometry test showed significant changes and the corresponding SAED patterns also exhibit polycrystalline and amorphous rings after HER test in alkaline electrolyte (Fig. 4b and 4c). Detailed TEM images taken under different reaction time conditions show that the nanosheets underwent a gradual amorphization process, particularly at the edges (Supplementary Fig. 25 and 26). It is also founded that the amorphization process would be more obvious at the thinner edges, which contribute to the formation of the functional amorphous layer for electrochemical reaction (Supplementary Fig. 27 and Supplementary Movie S2). According to some previous work, the amorphous layer can also protect the inner part of the nanosheet from over-etching, and effectively enhance the stability. We also conducted the in situ liquid TEM and ex situ TEM measurements to investigate the possible thickness of the Ru-NiPS₃ NSs, which demonstrate that the final thickness of the outer amorphous layer would be about 8 nm. (Supplementary Fig. 28, and Supplementary Movie S3).

Supplementary Fig. 28 was demonstrated as follows (Fig. R1-3):

Fig. R1-3. *Ex situ* TEM images of Ru-NiPS₃ NSs (dipping for 16h) and the estimated thickness of amorphous layer after stability tests for varying durations (scale bar: 5 nm).

4. Did the authors acquire DPs from the sample on GC?

Response: Thanks for the comment. As demonstrated in Fig. R1-2, we acquired the DPs of sample from GC, which demonstrated clearer TEM images and SAED patterns.

5. The author's claim of amorphization based on liquid cell data is, in fact, not self-consistent with their own *ex situ* analysis.

a. The AC-STEM shows that most of the NS remains crystalline and any amorphization is superficial.

b. This is also supported by the negligible difference powder diffraction patterns.

Other comments

Response: We thank the reviewer for the comments. For our Ru-NiPS₃ sample, there are two main reasons that limit the observation of the amorphization process, that is the thickness of the nanosheet and the reaction time. In previous version, we tried to obtain a faster and obvious variation of the structure of the nanosheets, and to satisfy the requirement, we chose a relatively thin sample located on the Pt electrode (like demonstrated in the previous manuscript). We agree the reviewer that it may bring two difficulties to our observations, one is the thinner nanosheet would be amorphized rapidly, another is the influence of H₂ bubbles may make the result implausible. After

careful consideration, in our revised version, the new *in situ* liquid TEM measurements were further modified and optimized.

Firstly, we selected a representative sample with a relatively thick thickness, which is close to the average thickness of the synthesized nanosheets of this work. Thus, the evolution of the structure is slower, which is conducive to long-term observation.

Secondly, the sample we chose was located on the GC electrode, which is far away from the Pt electrode and minimize the impact of liquid disturbance caused by the bubbles during the reaction process.

After several trials, we obtained the following sequential TEM images (Fig. R1-4) and the corresponding SAED patterns (Fig. R1-5). As demonstrated in the Fig. R1-4 and R1-5, the overall crystallinity is still good, and the edge positions have more amorphization, which is proved by the gradual formation of amorphous ring in SAED. We further collected SAED patterns of more edge positions, which all consistently indicated that, at the thinner regions, amorphization has already taken place in the initial stages of the reaction (Fig. R1-6).

Fig. R1-4. *In situ* TEM sequential images showing the evolution of the Ru-NiPS₃ NSs on GC electrode in 125 min.

Fig. R1-5. *In situ* sequential SAED pattern corresponding to R1-4, showing the evolution of the Ru-NiPS₃ NSs on GC electrode in 125 min.

Fig. R1-6. (a) TEM image of the initial Ru-NiPS₃ NSs without electrochemical reaction. (b) and (c) are the TEM image of the select area in (a) (red square) after 10 min and 45

min reaction, respectively (scale bar: 0.1 μm). (d) and (e) are the corresponding SAED patterns for (b) and (c), respectively (scale bar: 5 1/nm).

6. The authors should mention in the main text the average thickness of the NSs.

Response: We would like to express our gratitude to the reviewer for their important comments. We would like to clarify that we did calculate the average thickness of our sample based on the SEM image in Supplementary Fig. 4, which was found to be approximately 150 nm. We have included this information in the revised manuscript (line 99). To further confirm the thickness results with higher accuracy, we also used atomic force microscopy (AFM) to analyze the thickness of our sample in revised manuscript. Fig. R1-7 (also demonstrated as Supplementary Fig. 5 in the Supplementary Information) shows the thicknesses of several different regions, which were all found to be around 150 nm. This further validates the accuracy of our SEM analysis results.

Fig. R1-7. AFM image and corresponding height profiles

7. Line 110. The authors specify that that the d-spacings for are the (130) and (1-30) of NiPS₃.

Response: We thanks the reviewer for the comment. We indeed made mistakes on the crystal plane, which according to the HRTEM image and the corresponding FFT image (Fig. R1-8) should be (130) and ($\bar{1}$ 30), respectively. We apologize again for this mistake and have made the necessary corrections in the revised manuscript (Line 114).

Fig. R1-8 HRTEM image and the corresponding FFT pattern of the selected region in the left panel.

8. Why are the chronopotentiometry experiments (@100 mA and @5 nA) fixed at positive current densities? Shouldn't the currents?

Response: Thanks for comment. This two chronopotentiometry measurements are used to express different concepts. In the macroscopic three-electrode electrolyzer system, the directly measured quantity is indeed the current. But this is not a suitable parameter for comparing the catalytic performance in different electrocatalysts. For example, a current of 100 mA is much larger than 10 mA, but the former one may be obtained on a larger electrode. Thus, current density, which is calculated by dividing the current by the geometric area of the working electrode, is more reliable for comparison. So, in the realistic three-electrode electrolyzer system, we usually use current density (mA cm^{-2}) to demonstrate the electrocatalytic performance. However, in the *in-situ* liquid TEM cell, it is very hard to count the actual area or loading of catalyst falling on the electrodes. Considering that *in situ* liquid-phase TEM mainly studies the catalytic reaction mechanism and directly observes the reaction process, a simple current application is sufficient to initiate the reaction. That is why we use current density and current to describe the reaction condition in macroscopic three-electrode electrolyzer system and liquid-phase TEM measurements, respectively.

We also would like to clarify that we did not fix the parameter at a positive current density of 100 mA cm^{-2} and a current of 5 nA. It is evident that for HER, the current

measured during the reaction should be negative. The current densities and currents expressed in the paper are taken as absolute values, which is a common way to demonstrate HER current density (e.g., *Nat. Commun.* 10, 3899 (2019), *Energy Environ. Sci.*, 2019,12, 3522-3529, *Nano-Micro Lett.* 15, 120 (2023)). To eliminate any ambiguity, we have changed the current density to -100 mA cm^{-2} in Fig. 3g in the revised version. The modified figure is presented below (Fig. R1-9).

Fig. R1-9 (a) HER polarization curves for carbon cloth, NiPS₃ powder, NiPS₃ NSs, Ru-NiPS₃ NSs, and Pt/C measured in 1 M KOH with a scan rate of 2 mV s^{-1} . (b) Tafel slope of different catalysts obtained from the polarization curves in (a). (c) Exchange current density of different samples extrapolated linearly from the Tafel slope in (b). (d) Nyquist plots of NiPS₃ powder, NiPS₃ NSs, and Ru-NiPS₃ NS (inset shows the equivalent circuit diagram). (e) Scan rate dependence of the average capacitive currents for NiPS₃ NSs and Ru-NiPS₃ NSs. (f) Comparison of the overpotential and Tafel slope among our catalyst and other reported NiPS₃-based electrocatalysts for alkaline HER.

(g) Chronoamperometry curve test for Ru-NiPS₃ NSs at a fixed current density of -100 mA cm⁻²

9. Authors should also provide the chronopotentiometry data from the liquid cell holder.

Response: Thanks for the suggestion. In the revised version, we added the chronopotentiometry data as Supplementary Fig. 24, which replaced the initial Fig. in the original version to demonstrate more clear structure of the *in situ* liquid TEM chip.

Details are as follows (Fig. R1-10).

Fig. R1-10. The chronopotentiometry data obtained during *in situ* liquid TEM measurements. The accompanying inset illustrates the structure of the liquid TEM chip.

Reviewer #2 (Remarks to the Author):

In this manuscript, the authors synthesized a Ru-NiPS₃ nanosheets (Ru-NiPS₃ NSs) catalyst by three-step procedure for the hydrogen evolution reaction (HER). The samples were characterized by SEM, EDS, XPS, XRD and *in situ* TEM. The catalyst properties have been systematically discussed through electrochemical test and DFT, which well-demonstrates the importance of the highly active amorphous surfaces. Thus, publication this work on nature communications could be recommended after carefully addressing the following issues regarding material characterizations and proposed mechanism.

Response: We appreciate the reviewer's favorable feedback on our manuscript and have made extensive revisions based on suggestions.

1. Could the surface amorphization of the catalyst be controlled accurately through the method provided in this paper? The detailed experimental process should be provided for others to repeat.

Response: Thanks for the suggestion. In fact, the surface amorphized layer was formed gradually during the HER process. The process is hard to be accurately controlled at this stage. There are two reasons: 1) the nanosheets are grown on carbon cloth desultorily with hydrothermal method, thus, the contact with electrolyte may not be exactly the same. 2) as demonstrated in the SEM image, the large contact areas between all the nanosheets may hinder sufficient contact between the electrolyte and the contacted regions, thereby delaying the amorphization process.

However, the thickness of the amorphous layer would gradually stabilize. It has been proved that, the amorphous layer can also protect the inner part of the material, which ensures the stability of the structure (*ACS Catal.*, 2018, 8(5): 4091-4102, *Matter*, 2021, 4(9): 2850-2873.). Thus, when the thickness of the amorphous layer grown to a certain level, the inner part would be separated from the electrolyte, which in return stabilize the materials for long term use. As shown in Fig. R2-1, we further obtain the HRTEM images which demonstrated that the thickness of the amorphous layer in the Ru-NiPS₃ nanosheet remained stable during the long-term stability testing process. It should be

mentioned that the amorphous layer is generated gradually during the HER stability test. After the initial activation step is completed, the catalytic performance of the material is roughly determined. The increased thickness mainly results in better protection of the inner part, which ensures the structural stability of the nanosheet.

Fig. R2-1. TEM images of Ru-NiPS₃ NSs (dipping for 16h) and the estimated thickness of amorphous layer after undergoing stability tests for varying durations (scale bar: 5 nm).

We have also included more details about the preparation protocol for the electrode in the Methods section. The revised section is presented below (the part highlighted in yellow is the modified content):

Electrochemical measurements

All electrochemical measurements were conducted using a typical three-electrode cell with a CHI 760E electrochemical workstation (CH Instruments, Inc. Shanghai). The as-synthesized electrode, Hg/HgO electrode, and a graphite rod (Alfa Aesar, 99.9995%) were used as the working electrode, reference electrode, and counter electrode, respectively. Before the LSV test in 1 M KOH electrolyte, all electrodes were activated by the cyclic voltammetry (CV) technique for 100 cycles to obtain stable LSV curves. The scan rate is 50 mV s⁻¹, within the potential range from -0.8 V vs. Hg/HgO to -1.5 V vs. Hg/HgO, and the total activate time is about 1h. To avoid the influence of the Ru

species dissolved in the electrolyte during the activation process, the electrode was replaced immediately when the activation process is finished. LSV curves were then obtained with a scan rate of 2 mV s^{-1} . In this work, all potentials were converted to RHE with the equation $E_{\text{RHE}} = E_{\text{Hg/HgO}} + 0.098 \text{ V} + 0.059 \times \text{pH}$. EIS measurements were carried out within a frequency range of 10^6 Hz to 10^{-2} Hz , and the charge transfer resistance (R_{ct}) obtained by fitting the EIS data was used for the iR correction.

2. Does the samples immerse in RuCl_3 solution for different durations affect the loading amount of Ru in the nanosheets. Is the Ru-NiPS₃ with 0.81wt% content of Ru the best one after optimization? The related information should be provided.

Response: We thank the reviewer for the useful suggestion. Compared with other doping method, ion-exchange is a relatively slow process. However, this method is a simple and scalable technique, which could effectively reduce the waste of precious metals, and realize single atom doping.

Under the conditions of our laboratory, we selected 5 different dipping interval times to demonstrate our proposal (that is 0.5 h, 2 h, 4 h, 16 h, and 20 h). The HER activities of these five electrodes are shown in **Supplementary Fig. 16.**, and we also put it here as Fig. R2-2). As demonstrated, when the dipping time reached 16 h~20 h, the activities of the electrode are almost the same. Considering the time issue, we then choose 16 h as the representative sample fur further study, and the Ru doping amount of this sample is about 0.81%. We assume that after dipping 16 h~20 h, the ion-exchange process will reach equilibrium, and the catalytic performance remain unchanged like shown in Fig. R2-2.

With the growth of the immersion time, the activities of the electrode may be better due to more Ru was loaded into the NiPS₃ lattice. However, the current experiments can already prove the concept we are focusing on.

Fig. R2-2. Comparison of HER performance for different dipping time in RuCl_3 solution.

Moreover, we have conducted the ICP-OES for other samples, which demonstrated clearly that the doping amount of Ru is increased with the dipping duration. Detailed data are shown in Table R2-1. According to the ICP-OES results, a concentration of 0.8% is considered as the maximum limit for ion exchange doping of Ru atom into the NiP_3 NSs

Table R2-1 The ICP-OES results for other NiP_3 electrodes treated with RuCl_3 solution for varying durations

Dipping Duration	Element	Content (wt. %)
0.5 h	Ni	4.59
	P	5.00
	S	11.61
	Ru	0.16
2 h	Ni	4.31
	P	5.22
	S	12.68
	Ru	0.22
4 h	Ni	3.29
	P	4.70
	S	14.01
	Ru	0.36
20 h	Ni	5.77
	P	3.77
	S	11.70
	Ru	0.79

3. Why are the Ru atoms mainly distributed at the edge of the NSs, forming a Ru-enriched shell? What is the mechanism in forming such kind of structure?

Response: Thank you for your insightful comment. We will address the question from two different perspectives.

First, the formation of amorphous layer in alkaline media is inevitable during the HER process for most transition metal based electrocatalyst (e.g., *Angew. Chem. Int. Ed.* 2018, *Adv. Mater.*, 2021, 33(45): 2103812.). However, the formed amorphous layer could provide effective protection for interior electrodes (*Matter*, 2021, 4(9): 2850-2873), and with the reaction progressing, the amorphous layer would protect the inner part from contacting of the electrolyte, which would stabilize the core-shell structure. Thus, the reaction primarily occurs at the outer amorphous layer, making it a key factor in determining the catalytic activity and efficiency of the electrocatalyst, and the modification (such as increase the specific surface area, increase the number of active sites, and so on) is mainly focused on the amorphous layer.

Secondly, the amorphous layer contains many bridging S_2^{2-} species, and many dangling bonds, which contribute to the adsorption of Ru species. Our DFT calculation results also indicated that Ru bonded with S_2^{2-} species is more stable than other possible position (Fig. R2-3, also demonstrated in Supplementary Table 5), and tend to form stable Ru rich edges for reaction.

Fig. R2-3. DFT calculation results of adsorption energy for different Ru doping sites

4. The authors demonstrated that the Ru⁰ or Ru cluster was unstable during the HER test, and only Ru⁴⁺ species remained at the edge sites of NiPS₃ NSs (Fig. 5j). So, are these Ru⁰ or Ru cluster species converted to Ru⁴⁺ or dissolved in the electrolyte? Further experimental verification is suggested.

Response: We thank the reviewer for the comment. To verify this, we further measured the ICP-OES of the electrolyte after different reaction duration. As shown in Fig. R2-4, the content of Ru species remained almost unchanged after test for 2 h, which proved the unstable Ru⁰ species would dissolve in the electrolyte. The Fig. R2-4 is also included in the Supplementary Information as Supplementary Fig.30.

Fig. R2-4. ICP-OES data of Ru species in electrolyte after different reaction time.

The manuscript is also modified: Line 293 “*Further ICP-OES measurements confirmed that the Ru species gradually dissolved into the electrolyte, and the remaining Ru⁴⁺ species served as the active species for the HER process (Supplementary Fig. 30)*”

5. Is all Ru scattered around the edges after electrolysis? Does the amorphous state form only on the [001] crystal plane (Fig. 5a)? Or are there amorphous states in other crystal

planes? It is recommended to supplement other regional HAADF images.

Response: We appreciate the reviewer's comment. The Ru atoms were randomly doped into the NiPS₃ lattice, which means that they may exist both at the edges and in the inner part of the nanosheets. However, during the self-reconstruction process, Ru atoms may gradually leach out like other metal cations. The amorphous layer, therefore, would serve as an absorbent for Ru atoms to reduce the leaching of Ru. Although Ru atoms may exist in the inner crystalline part, the HER process would only occur at the surface or edges of the nanosheets. This means that only the Ru atoms at the amorphous layer can contact with the electrolyte and serve as the active sites for HER.

In fact, the amorphization process did not depend on the specific crystal plane, and the edges of the nanosheets may undergo a certain degree of amorphization process. We further acquired the AC-HAADF STEM image from other regions to demonstrate this. The details are shown in Fig. R2-5 (also added into the Supplementary Information as Supplementary Fig. 29).

Fig. R2-5. HAADF-STEM images of Ru-NiPS₃ along [001] and [103] zone axis, which showed that the amorphization process is independent of the crystal plane orientation.

6. From Fig. 4 it is hesitant to conclude anything from the *in situ* electrochemical liquid

cell TEM holder. The TEM images and the contrast are not convincing enough.

Response: We are grateful to the reviewer for drawing our attention to the shortcomings in our previous *in-situ* TEM experiments. After careful consideration and analysis of the possible issues, we conducted new, additional *in-situ* TEM experiments, and the results are presented as Fig. R2-6, and we also replaced the initial version of Fig. 4 with the revised figure.

Fig. R2-6. (a) Schematic illustration of the *in situ* electrochemical liquid cell TEM holder and the liquid cell. *In situ* liquid TEM image (scale bar: 0.2 μm) of Ru-NiPS₃ NSs (b) before and (c) after chronopotentiometry test. (d) and (e) demonstrated the corresponding SAED patterns (scale bar: 5 1/nm) for (b) and (c), respectively.

The description of Fig. 4 is also modified: From Line 246

“Observation and characterization of surface reconstruction process. *In situ liquid electrochemical TEM technique was conducted to gain insights into the structural evolution of Ru-NiPS₃ NSs during the alkaline HER process (Fig. 4a demonstrated the structure of the in situ electrochemical liquid cell TEM holder and the liquid cell). To minimize interference from other factors in the analysis, we first ruled out two potential sources of disturbance: corrosion of the sample by the alkaline solution, and interference from the electron beam on the sample. The Ru-NiPS₃ NSs electrode was first immersed into 1 M KOH for 24 h to show the influence of the alkaline electrolyte. As demonstrated in Supplementary Fig. 23a, the morphology demonstrated negligible variation after the immersing process, which is also proved by the corresponding XRD pattern (Supplementary Fig. 23b) and Raman spectra (Supplementary Fig. 23c). To rule out the influence of the electron beam in the TEM, the assembled liquid in situ electrochemical TEM cell was placed into the TEM, and upon in situ irradiation, no significant changes to the sample were observed (Supplementary Movie S1). The above result demonstrated that both alkaline solution and electron beam in TEM have little influence on the sample. The in situ liquid electrochemical TEM measurement was conducted under a constant current of -5 nA vs. Pt (Supplementary Fig. 24). The in situ TEM images before and after 2 h continually chronopotentiometry test showed significant changes and the corresponding SAED patterns also exhibit polycrystalline and amorphous rings after HER test in alkaline electrolyte (Fig. 4b and 4c). Detailed TEM images taken under different reaction time conditions show that the nanosheets underwent a gradual amorphization process, particularly at the edges (Supplementary*

Fig. 25 and 26). It is also founded that the amorphization process would be more obvious at the thinner edges, which contribute to the formation of the functional amorphous layer for electrochemical reaction (Supplementary Fig. 27 and Supplementary Movie S2). According to some previous work, the amorphous layer can also protect the inner part of the nanosheet from over-etching, and effectively enhance the stability. We also conducted the *in situ* liquid TEM and *ex situ* TEM measurements to investigate the possible thickness of the Ru-NiPS₃ NSs, which demonstrate that the final thickness of the outer amorphous layer would be about 8 nm. (Supplementary Fig. 28, and Supplementary Movie S3).”

From *in situ* liquid-phase transmission electron microscopy, we can draw the following conclusions:

(1) The *in-situ* liquid TEM sample are demonstrated as Fig. R2-7a. And the morphology variation of the edge site (orange square) is demonstrated in Fig. R2-7b, which shows significant amorphization process, especially at the edges. Relative thin nanosheets will be amorphized faster (red square, Fig. R2-7c). Thicker regions, on the other hand, the crystallinity is better maintained, which further proved the thinner edges at the nanosheets would be easily amorphized.

Fig. R2-7. (a) TEM image of the selected sample for *in situ* liquid TEM measurement. (b) *In situ* TEM sequential images showing the evolution of the Ru-NiPS₃ NSs on GC electrode in 125 min. (c) SAED patterns of the select area in (a) (red square) after 10 min and 45 min reaction, respectively (scale bar: 5 1/nm)

2) The main part of the sample remained relatively stable throughout the reaction process according to the SAED pattern (Fig. R2-8). However, when the reaction reached about 75 min, we can also observe the formation of polycrystal rings and amorphous rings generated with the reaction going on. This result further confirmed the amorphous process will mainly happen on the edges or surfaces without destroying the structure of nanosheets, which is also proved by the XRD and XPS results obtained after stability test (Fig. 5).

Fig. R2-8. *In situ* sequential SAED pattern showing the evolution of the Ru-NiPS₃ NSs

7. In the XPS, the same element fitting should be redone with all components having the same FWHM. In Fig. 5 should be refitted it.

Response: We thank the reviewer for the comment. According to the reviewer's suggestion, we refitted the XPS data in Fig. 5. In the revised figure, we keep the FWHM

of the fitting components close to each other in the same element. Details are as follows (Fig. R2-9).

Fig. R2-9. The revised XPS spectra for (a) Ni $2p_{3/2}$, (b) P $2p$, (c) S $2p$, and (d) Ru $3p_{3/2}$ of Ru-NiPS3 NSs after HER test.

We also changed the description of in the manuscript about the XPS result of the S $2p$ spectra, since the broad peak located around 170 eV was deconvoluted into two different peaks which belongs to the sulfate species $[\text{SO}_4]^{2-}$ (at ~ 170.8 eV) and sulfite $[\text{SO}_3]^{2-}$ (at ~ 168.5 eV)

The description of Fig.5i is also modified: From Line 351

“Another broad peak at ~ 169.6 eV was deconvoluted into two different peaks which belongs to the sulfate species $[\text{SO}_4]^{2-}$ (at ~ 170.8 eV) and sulfite species $[\text{SO}_3]^{2-}$ (at ~ 168.5 eV) (Fig. 5i).”

And the revised Fig. 5 in the main text are as follows (Fig. R2-10):

Fig. R2-10. (a) HAADF-STEM image of Ru-NiPS₃ after long-term stability HER test in 1 M KOH. (b), (c) the enlarged part of amorphous part and inner part of Ru-NiPS₃ from (a). (d) EDS mapping of Ni, P, S, and Ru element (scale bar: 2 nm). (e) PXRD image of Ru-NiPS₃ NSs after HER test. (f) Raman spectrum of Ru-NiPS₃ NSs after HER test. (g), (h), (i), (j) XPS spectra for Ni 2p_{3/2}, P 2p, S 2p, and Ru 3p_{3/2} of Ru-NiPS₃ NSs after HER test.

Reviewer #4 (Remarks to the Author):

In this work, the authors aim to address a commonly overlooked issue in the alkaline HER: the formation of an amorphous layer during the reaction process. The lack of evidence for the in-situ characterization of the amorphous layer formed during the reaction has hindered the full development of its essential role and potential applications. Fu et al. have utilized an advanced in-situ liquid TEM technique to directly demonstrate the formation of the amorphous layer, which provided direct evidence of the amorphization. Their results prove that the amorphous layer is significant in analyzing the true catalytic mechanism and active sites for the HER process. Furthermore, the authors have also shown, both theoretically and experimentally, that by rationally designing and utilizing the unavoidable amorphous layer, the catalytic performance of the electrocatalyst can be significantly improved. I highly recommend this manuscript for publication in Nature Communications. Please find below some detailed comments for the authors to consider.

Response: We thank reviewer for his positive comments on our manuscript.

1. The manuscript was written mainly to discuss the amorphous layer, will the sample be oxidized during the storage? And if the sample was oxidized after the preparation, will the oxidized layer have any influence on the catalytic process?

Response: Thanks for the comment. Indeed, the oxidation of transitional metal based electrocatalysts is sometimes inevitable in the atmosphere, which has been already recognized by some previous works, such as *Angew. Chem. Int. Ed.*, 54: 14710-14714., *Nat Commun* 7, 13216 (2016), *Nat. Energy* 6, 1144–1153 (2021), *Nat Commun.* 12, 3540 (2021), and so on. This phenomenon also suits the situation in our case, which could be confirmed by the HRTEM images in Fig. 1b and Supplementary Fig. 7. It can also be seen that the thickness edges of the NiPS₃ and Ru-NiPS₃ NSs are similar (less than 1 nm). Therefore, if any influence of inevitable oxidation occurred due to exposure to air, it should be similar for both samples.

Despite this possibility, we observed an obvious increase in HER activity. Based on this evidence, we can conclude that the enhancement is due to the proposed Ru-enriched

edge sites, rather than the slight oxide layer that may have formed. Additionally, we have included more HRTEM images of the Ru-NiPS₃ samples below, which demonstrate that any surface oxidation present is minimal. Details are as follows:

Fig. R4-1. HRTEM image of other three different samples of Ru-NiPS₃, which demonstrated the oxidation layer are similar (scale bar: 1 nm).

2. As shown in Fig. 2g, some Ru(0) was detected before the HER test. I think this should belong to the Ru cluster or Ru metal. Will the Ru(0) have any influence on the catalytic performance?

Response: We thank the reviewer for the comments. Based on the XPS analysis, we found that Ru(0) was present in the initial sample, but disappeared after stability testing. To further investigate the behavior of elements during the HER process, we conducted ICP-OES testing of the electrolyte before and after the HER test. The results showed that the content of Ru increased and eventually reached a constant level as the reaction progressed. This suggests that the unstable Ru(0) species were removed from the NiPS₃ NSs, while the Ru that substituted in the NiPS₃ lattice remained stable and served as the active sites for the HER reaction. We have included the ICP-OES results as Supplementary Fig. 30.

Fig. R4-2. ICP-OES data of Ru species in electrolyte after different reaction time.

3. The in-situ TEM images and movies have demonstrated that the NiPS₃ NSs undergo an amorphization process. Is it possible that the electrolyte could have an impact on the formation of the amorphous layer on the catalyst surface?

Response: Thanks for the insightful comments. We conducted an additional experiment to investigate the impact of alkaline media on Ru-NiPS₃ samples. Specifically, we immersed another Ru-NiPS₃ sample in a 1 M KOH solution for 24 hours. As shown in Fig. R4-3(a), we observed that the edges of the nanosheet remained relatively unchanged after immersion, suggesting that the nanosheet is stable under alkaline conditions. Furthermore, XRD pattern (Fig. R4-3(b)) and Raman spectroscopy (Fig. R4-3(c)) analysis provided additional support for our findings, indicating that the alkaline solution did not significantly alter the structure of Ru-NiPS₃ NSs. These results support the conclusion that the amorphous layer observed during the HER test is likely due to in-situ reconstruction of the Ru-NiPS₃ NSs, rather than a result of simple immersion. The results also included in the Supplementary Information as Supplementary Fig. 23.

Fig. R4-3. (a) HRTEM image, (b) XRD pattern, and (c) Raman spectrum of Ru-NiPS₃ NSs after immersing in 1 M KOH solution for 24h.

4. In Fig. 1e, the author used a CIF profile to demonstrate the crystal structure of NiPS₃. It would be better if changed it into a HAADF-STEM simulation image.

Response: We would like to express our gratitude to the reviewer for their valuable suggestion. As per their recommendation, we have simulated the HAADF-STEM image with Dr. Probe software in the revised version. The modified Fig. 1 is presented as Fig. R4-4:

Fig. R4-4. (a) SEM image of the as-prepared Ru-NiPS₃ NSs. (b), (c) HRTEM image and the corresponding SAED image along the [001] zone axis. (d) Atomic-level HAADF-STEM image of an ultrathin NiPS₃ nanosheet. (e) Crystal structure of NiPS₃ along [001] zone axis. (f) Line-scanning intensity profile obtained from the selected area in (d). (g) and (h) are the simulated HAADF-STEM image with Ru doped into the NiPS₃ lattice and the corresponding line-scanning intensity profile which is similar to the experimental one. (i) HEDF-STEM image corresponding EDS mapping of Ru-NiPS₃ nanosheet (scale bar 500 nm).

5. Some typos in the manuscript should be corrected. For instance, in line 57, the word “lead” should be “leads”. In line 115” characterization is conducted” should be “characterization was conducted”. In line 355, it is recommended to spell out the abbreviation "DI" in its complete form, when it is used for the first time in a document.

Response: Thanks for the comments. We have carefully reviewed our work and made further revisions to address the typos and grammar mistakes, including those that the reviewer pointed out.

REVIEWER COMMENTS

Reviewer #1 (Remarks to the Author):

I will again mainly comment on the electron microscopy work. In this revision, the authors made a good effort at improving the quality of the electron microscopy results and the liquid cell TEM data is now more convincing. I have a few more comments for the authors.

1. The evolution of the diffraction patterns in Supplementary Figure 26 is interesting and suggests a re-orientation of the nanosheets during the experiment. However, the authors barely discussed this aspect. What lattice spacings/structures do points depicted by the curve line, the sharp spots just within the curve line, and the ring of diffraction nearer to the transmitted beam correspond to? These should be labeled in Figure 4e.

2. It will be inaccurate to describe the sample after 125 min as a polycrystal as the term implies a randomly oriented structure. The diffraction pattern in 4e clearly shows that the structure is not random and that clearly orientational relationships within the larger nanosheet flake. I suggest that the authors revisit this section and update the discussion accordingly.

3. Are the authors able to determine the amorphous layer thickness from the liquid TEM experiments or at least make it clearer in the images? How does the amorphous layer thickness from the liquid TEM experiments that compare with the ex situ results after similar times?

4. It is also not explained sufficiently what Supplementary Figure 28 is about (Page 11, 261-264). The Figure caption only mentions dipping, but no information about the applied electrochemical conditions and how the samples were obtained.

5. Response to my previous comment 2(b): "What should be mentioned is that, we adopted a currently commonly used "thin liquid strategy" to minimize the influence of liquid on the resolution of TEM, so that we can acquire some sufficiently clear TEM images during our experiments."

Just because an approach has been used in prior work does not mean that does not have its issues. For example, early work in the field of liquid cell TEM commonly used electron dose rates of hundreds to even thousands of electrons/A²/s but now, we know that such dose rates are not tenable for realistic experiments. Here, I suggest that the authors rephrase the sentence in the methods (page 16, line 406-408) to something similar to what I have outlined below.

"For these in situ liquid TEM experiments, the samples were imaged within a thin liquid layer so that we can acquire sufficiently clear TEM images with good spatial resolution. In this case, one should note that the exact applied conditions are not identical to that of a realistic electrochemical cell, which may lead to differences between the results obtained from in situ and ex situ measurements.

Reviewer #2 (Remarks to the Author):

In this manuscript, the authors applied trace Ru doped NiPS₃ nanosheets for high-efficiency alkaline hydrogen evolution reaction. Catalyst properties have been systematically discussed through electrochemical test and compared experiments without obvious errors. Stability tests are specific and persuasive. In particular, the feedback on the in situ TEM section is very detailed and easier to understand. I think the quality of this manuscript has been enhanced considerably after the revision. After addressing some points, this his manuscript can be considered for publication in Nat. Commun.

1. The XPS survey spectrum for Ru-NiPS₃ NSs should be provided.

2. Some important works in this field should be added, such as: DOI:10.1016/j.nanoen.2020.105375. DOI: 10.1002/sml.201902427.

Reviewer #4 (Remarks to the Author):

After review the response from the author, I recommend the manuscript can be accepted for publication at current status.

REVIEWER COMMENTS

Reviewer #1 (Remarks to the Author):

I will again mainly comment on the electron microscopy work. In this revision, the authors made a good effort at improving the quality of the electron microscopy results and the liquid cell TEM data is now more convincing. I have a few more comments for the authors.

1. The evolution of the diffraction patterns in Supplementary Figure 26 is interesting and suggests a re-orientation of the nanosheets during the experiment. However, the authors barely discussed this aspect. What lattice spacings/structures do points depicted by the curve line, the sharp spots just within the curve line, and the ring of diffraction nearer to the transmitted beam correspond to? These should be labeled in Figure 4e.

2. It will be inaccurate to describe the sample after 125 min as a polycrystal as the term implies a randomly oriented structure. The diffraction pattern in 4e clearly shows that the structure is not random and that clearly orientational relationships within the larger nanosheet flake. I suggest that the authors revisit this section and update the discussion accordingly.

3. Are the authors able to determine the amorphous layer thickness from the liquid TEM experiments or at least make it clearer in the images? How does the amorphous layer thickness from the liquid TEM experiments that compare with the ex situ results after similar times?

4. It is also not explained sufficiently what Supplementary Figure 28 is about (Page 11, 261-264). The Figure caption only mentions dipping, but no information about the applied electrochemical conditions and how the samples were obtained.

5. Response to my previous comment 2(b): "What should be mentioned is that, we adopted a currently commonly used "thin liquid strategy" to minimize the influence of liquid on the resolution of TEM, so that we can acquire some sufficiently clear TEM images during our experiments."

Just because an approach has been used in prior work does not mean that does not have its issues. For example, early work in the field of liquid cell TEM commonly used electron dose rates of hundreds to even thousands of electrons/A²/s but now, we know that such dose rates are not tenable for realistic experiments. Here, I suggest that the authors rephrase the sentence in the methods (page 16, line 406-408) to something similar to what I have outlined below.

"For these in situ liquid TEM experiments, the samples were imaged within a thin liquid layer so that we can acquire sufficiently clear TEM images with good spatial resolution.

In this case, one should note that the exact applied conditions are not identical to that of a realistic electrochemical cell, which may lead to differences between the results obtained from in situ and ex situ measurements.

Reviewer #2 (Remarks to the Author):

In this manuscript, the authors applied trace Ru doped NiPS₃ nanosheets for high-efficiency alkaline hydrogen evolution reaction. Catalyst properties have been systematically discussed through electrochemical test and compared experiments without obvious errors. Stability tests are specific and persuasive. In particular, the feedback on the in situ TEM section is very detailed and easier to understand. I think the quality of this manuscript has been enhanced considerably after the revision. After addressing some points, this his manuscript can be considered for publication in Nat. Commun.

1. The XPS survey spectrum for Ru-NiPS₃ NSs should be provided.
2. Some important works in this field should be added, such as: DOI:10.1016/j.nanoen.2020.105375. DOI: 10.1002/sml.201902427.

Reviewer #4 (Remarks to the Author):

After review the response from the author, I recommend the manuscript can be accepted for publication at current status.

RESPONSE TO REVIEWER COMMENTS

Title: “Unraveling and Leveraging *in situ* Surface Amorphization for Enhanced Hydrogen Evolution Reaction in Alkaline Media”

(Manuscript ID: NCOMMS-23-21960A)

We are grateful to all the reviewers for their affirmation of our work and valuable suggestions. Based on the feedback from Reviewer 1 and Reviewer 2, we have made further revisions to the manuscript in order to meet the publication requirements of *Nature Communications*. A point-by-point response to all comments is given below, and our responses are in blue text. The corresponding modifications are also given in red text in the *Manuscript* and *Supplementary Information*.

Reviewer #1 (Remarks to the Author):

I will again mainly comment on the electron microscopy work. In this revision, the authors made a good effort at improving the quality of the electron microscopy results and the liquid cell TEM data is now more convincing. I have a few more comments for the authors.

Response: Thank you for recognizing our efforts. We will now proceed to address your comments one by one.

1. The evolution of the diffraction patterns in Supplementary Figure 26 is interesting and suggests a re-orientation of the nanosheets during the experiment. However, the authors barely discussed this aspect. What lattice spacings/structures do points depicted by the curve line, the sharp spots just within the curve line, and the ring of diffraction nearer to the transmitted beam correspond to? These should be labeled in Figure 4e.

Response: Thanks for your insightful comment. Just like the reviewer's comment, the *in situ* liquid TEM image demonstrated in manuscript Fig. 4 and Supplementary Figure 26 indicated a re-orientation of the nanosheets during the experiment, and this is also the information we wanted to convey, that the reconstruction process of the catalyst is a gradual evolution accompanied by certain changes in the crystal structure. This is one

of the advantages of *in situ* liquid TEM technique. According to the reviewers' suggestions, we have further modified Figure 4e, and labeled the corresponding lattice spacing, sharp spots, and the ring of diffraction nearer the transmitted beam. Details are as follows:

Figure R1-1 (a) Schematic illustration of the *in situ* electrochemical liquid cell TEM holder and the liquid cell. *In situ* liquid TEM image (scale bar: 0.2 μm) of Ru-NiPS₃ NSs (b) before and (c) after chronopotentiometry test. (d) and (e) demonstrated the corresponding SAED patterns (scale bar: 5 1/nm) for (b) and (c), respectively. The *in situ* TEM image provided clear evidence of a reconstruction process occurring at the edges of the NSs during the HER process. Additionally, the corresponding SAED pattern confirmed that a portion of the NSs underwent a transformation into a polycrystalline and amorphous state

2. It will be inaccurate to describe the sample after 125 min as a polycrystal as the term implies a randomly oriented structure. The diffraction pattern in 4e clearly shows that the structure is not random and that clearly orientational relationships within the larger nanosheet flake. I suggest that the authors revisit this section and update the discussion accordingly.

Response: Thanks for your comments. We agree with the reviewer that even after 125 min test, the nanosheet still mainly keep the orientational relationships within the larger nanosheet flake. In our previous description, there were inaccuracies that led to misunderstandings. We would like to clarify our original intention, which was to convey that during the reaction process, there is a localized amorphization and polycrystallization at the edges of the nanosheets. However, it is important to note that the overall crystallinity of the nanosheets remains largely unaffected, as evidenced by the distinct crystalline nature observed in the SAED patterns. We apologize for any confusion caused by our previous wording. To eliminate any ambiguity in the description, we have made the following modifications:

(Page10, Line 252-258) *“After continuously subjecting the Ru-NiPS₃ NSs to a 2-hour chronopotentiometry test, a significant reconstruction was observed at the edge position of the NSs (Figure 4b and 4c). The corresponding selected area electron diffraction (SAED) patterns clearly showed that while most of the NSs remained unchanged after the chronopotentiometry test (with similar diffraction spots as in Figure 4d), a portion of the nanosheet underwent a transformation into a polycrystalline or amorphous state during the reconstruction process (as indicated by the presence of faint polycrystalline rings and amorphous halo ring in Figure 4e).”*

3. Are the authors able to determine the amorphous layer thickness from the liquid TEM experiments or at least make it clearer in the images? How does the amorphous layer thickness from the liquid TEM experiments that compare with the ex situ results after similar times?

Response: Thanks for your valuable comments. In our in situ TEM experiments, due to the limitations of resolution, we are unable to obtain clear images like those from

non-in situ transmission electron microscopy (TEM) images. In the paper we have presented the clearest result obtained after multiple attempts. As mentioned by the reviewer, it is indeed important to clearly mark the approximate locations of the amorphous regions or areas of significant reconstruction in the images. Thus, we further marked the places and showed the approximate extent of amorphization in Figure 4c (as demonstrated in Figure R1-1).

As for the comparison with the ex situ results after similar times, it may indeed be inappropriate to directly compare in situ liquid-phase TEM images with non-in situ high-resolution TEM (HRTEM) images. This is because these two techniques differ in sample preparation and environmental conditions.

In situ liquid-phase TEM is typically used to study the structure and dynamic behavior of materials in liquid environments. It provides information about materials under in situ conditions, but image quality may be limited due to factors such as resolution and contrast, which can be influenced by the presence of the liquid environment.

On the other hand, ex situ HRTEM is commonly used to investigate the crystal structure and details of materials. It is conducted under vacuum or dry conditions and can achieve higher resolution and clearer images.

As shown in Figure 4 and Supplementary Fig. 26 and Fig.27, The in situ TEM images obtained in liquid phase using our in situ facility can only achieve resolutions on the scale of few tens of nanometers, which is insufficient for clearly studying the thickness of amorphous layers (which, as revealed by ex situ HRTEM image, is less than 10 nm). Considering the significant differences between the testing conditions in in situ liquid-phase TEM and the actual electrolytic cell reactions, we are inclined to utilize in situ liquid-phase TEM for investigating the overall structural evolution of the nanosheets, while employing non-in situ TEM techniques for more detailed examination of localized reconstruction or amorphization phenomena.

4. It is also not explained sufficiently what Supplementary Figure 28 is about (Page 11, 261-264). The Figure caption only mentions dipping, but no information about the applied electrochemical conditions and how the samples were obtained.

Response: Thank you for the reviewer's suggestions, and we sincerely apologize for the confusion caused. In Figure 28, we present the changes in the thickness of the edge amorphous layer of Ru-NiPS₃ nanosheets with increasing HER reaction time. The label 'dipping for 16h' following Ru-NiPS₃ in the caption refers to the sample that underwent ion exchange for 16 hours (details are demonstrated in *Supplementary Fig.13-Fig. 14, Fig. 16-Fig. 17*), representing the typical sample as described in our paper. It is important to note that the 'dipping for 16h' label in the caption does not correspond to the duration of the electrochemical reaction depicted in the figure. We apologize for any confusion caused by this discrepancy.

To eliminate any ambiguity, we have made the following modifications to the main text and the caption of Supplementary Fig. 28 (now Supplementary Fig. 29 in the revised version):

(Page 11, Line 256-273) *“Ex situ HRTEM images of Ru-NiPS₃ after stability test for different reaction duration (from 1 h to 16 h) are demonstrated in Supplementary Fig. 29, which showed that with the reaction time increases, the thickness of the in situ formed amorphous layer gradually increases, eventually reaching a roughly stable thickness (~ 7.5 nm for 16 h). According to some previous work, the amorphous layer can also protect the inner part of the nanosheet from over-etching, and effectively enhance the stability, which is proved by the in situ liquid TEM, which demonstrated that after the amorphization process is complete, the morphology and edges of the material undergo minimal observable changes (Supplementary Movie S3).”*

Furthermore, we rewrite the caption of Supplementary Fig. 28 and added necessary experimental details to eliminate the disambiguation (NOTE: since a survey XPS data was added into the Supplementary Information, the figure has now become Supplementary Fig. 29 in the revised version). The revised Figure and the caption are demonstrated as Figure R1-2.

Figure R1-2. *Ex situ* TEM images of the representative sample Ru-NiPS₃ NSs and the estimated thickness of amorphous layer after stability tests for varying durations (1 h, 4 h, 10 h, and 16h reaction duration; scale bar: 5 nm). The electrochemical test was operated in 1 KOH solution, with a constant current density of -100 mA cm⁻². The results demonstrated that as the reaction progresses, the thickness of the amorphous layer around the nano flakes tends to stabilize (~8 nm), thereby stabilizing the overall structure of the catalyst.

5. Response to my previous comment 2(b): "What should be mentioned is that, we adopted a currently commonly used "thin liquid strategy" to minimize the influence of liquid on the resolution of TEM, so that we can acquire some sufficiently clear TEM images during our experiments."

Just because an approach has been used in prior work does not mean that does not have its issues. For example, early work in the field of liquid cell TEM commonly used electron dose rates of hundreds to even thousands of electrons/A²/s but now, we know that such dose rates are not tenable for realistic experiments. Here, I suggest that the authors rephrase the sentence in the methods (page 16, line 406-408) to something similar to what I have outlined below.

"For these in situ liquid TEM experiments, the samples were imaged within a thin liquid layer so that we can acquire sufficiently clear TEM images with good spatial resolution.

In this case, one should note that the exact applied conditions are not identical to that of a realistic electrochemical cell, which may lead to differences between the results obtained from in situ and ex situ measurements.

Response: Thank you for providing us with such detailed and constructive feedback. We completely agree with your point that even commonly used techniques can have certain issues. Therefore, we cannot use this as a justification for describing the selection of experimental conditions. Under the circumstances of our paper, a more appropriate approach to describing the experimental conditions would be to provide an objective description rather than basing it solely on the fact that the method has been used before. Based on the template you provided, we have made the following modifications to the relevant section:

(Page 17, Line 415-419) *“In order to obtain clear TEM images with good spatial resolution, the samples in these in situ liquid TEM experiments were imaged within a thin liquid layer. It should be noted that the applied experimental conditions in these studies may not perfectly replicate those of a realistic electrochemical cell. As a result, there may be differences between the results obtained from in situ and ex situ measurements.”*

We would like to express our heartfelt gratitude once again for the meticulous and detailed references provided by the reviewer.

Reviewer #2 (Remarks to the Author):

In this manuscript, the authors applied trace Ru doped NiPS₃ nanosheets for high-efficiency alkaline hydrogen evolution reaction. Catalyst properties have been systematically discussed through electrochemical test and compared experiments without obvious errors. Stability tests are specific and persuasive. In particular, the feedback on the in situ TEM section is very detailed and easier to understand. I think the quality of this manuscript has been enhanced considerably after the revision. After addressing some points, this his manuscript can be considered for publication in Nat. Commun.

Response: We greatly appreciate the recognition and constructive feedback provided by the reviewer on our work.

1. The XPS survey spectrum for Ru-NiPS₃ NSs should be provided.

Response: Thanks for your suggestion. We have added the XPS survey spectrum of Ru-NiPS₃ NSs as demonstrated in Fig. R2-1 (also demonstrated as Supplementary Fig. 15 in the revised version).

Fig. R2-1 XPS survey spectrum for Ru-NiPS₃ NSs

2. Some important works in this field should be added, such as:

DOI:10.1016/j.nanoen.2020.105375. DOI: 10.1002/sml.201902427.

Response: Thanks for the suggestion. After careful review, we believe that the references you provided are highly relevant to our work. In the revised manuscript, they have been included as references 13 and 28 in the manuscript, respectively.

13. Li, X. et al. High-yield electrochemical production of large-sized and thinly layered NiPS₃ flakes for overall water splitting. *Small* **15**, 1902427 (2019).

28. Lu, S. et al. Underwater superaerophobic Ni nanoparticle-decorated nickel–molybdenum nitride nanowire arrays for hydrogen evolution in neutral media. *Nano Energy* **78**, 105375 (2020)

Reviewer #4 (Remarks to the Author):

After review the response from the author, I recommend the manuscript can be accepted for publication at current status.

Response: We are very appreciated for your approval on our revised manuscript.

REVIEWERS' COMMENTS

Reviewer #1 (Remarks to the Author):

The authors have adequately addressed my comments. I recommend the current manuscript for publication.